# Rethinking Neural Operations for Diverse Tasks

**Nicholas Roberts**[*]
University of Wisconsin-Madison
nick11roberts@cs.wisc.edu

**Mikhail Khodak**[*]
Carnegie Mellon University
khodak@cmu.edu

**Tri Dao**
Stanford University
trid@stanford.edu

**Liam Li**
Hewlett Packard Enterprise
me@liamcli.com

**Christopher Ré**
Stanford University
chrismre@cs.stanford.edu

**Ameet Talwalkar**
Carnegie Mellon University & Hewlett Packard Enterprise
talwalkar@cmu.edu

## Abstract

An important goal of AutoML is to automate-away the design of neural networks on new tasks in under-explored domains. Motivated by this goal, we study the problem of enabling users to discover the right neural operations given data from their specific domain. We introduce a search space of operations called XD-Operations that mimic the inductive bias of standard multi-channel convolutions while being much more expressive: we prove that it includes many named operations across multiple application areas. Starting with any standard backbone such as ResNet, we show how to transform it into a search space over XD-operations and how to traverse the space using a simple weight-sharing scheme. On a diverse set of tasks—solving PDEs, distance prediction for protein folding, and music modeling—our approach consistently yields models with lower error than baseline networks and often even lower error than expert-designed domain-specific approaches.

## 1 Introduction

Automated machine learning (AutoML) and neural architecture search (NAS) are often motivated by a vision of democratizing ML by reducing the need for expert design on a variety of tasks. While NAS has grown rapidly with developments such as weight-sharing [36] and "NAS-benches" [47, 49], most efforts focus on search spaces that glue together established primitives for well-studied tasks like vision and text [32, 26, 45, 25] or on issues such as latency [8, 13]. In this work, we revisit the broader vision of NAS and propose to move towards much more general search spaces while still exploiting successful network topologies. To do so we focus on expanding the set of operations, which is usually fairly small; for example, that of the well-studied DARTS space has eight elements: a few types of convolution and pooling layers [32]. The baseline approach for expanding this set—adding operations one-by-one—scales poorly and will not result in new operations when faced with new types of data.

Our core contribution is a re-imagining of NAS operation spaces that drastically expands this set in a principled fashion to include both standard operations as well as a wide range of new ones. To do so we exploit the fact that most standard operations used in modern NAS return linear transforms diagonalized by the discrete Fourier transform (DFT). Replacing the DFT matrices in the diagonal decomposition by a more expressive family of efficient linear transforms known as *Kaleidoscope* or

---

    * denotes equal contribution.

35th Conference on Neural Information Processing Systems (NeurIPS 2021).

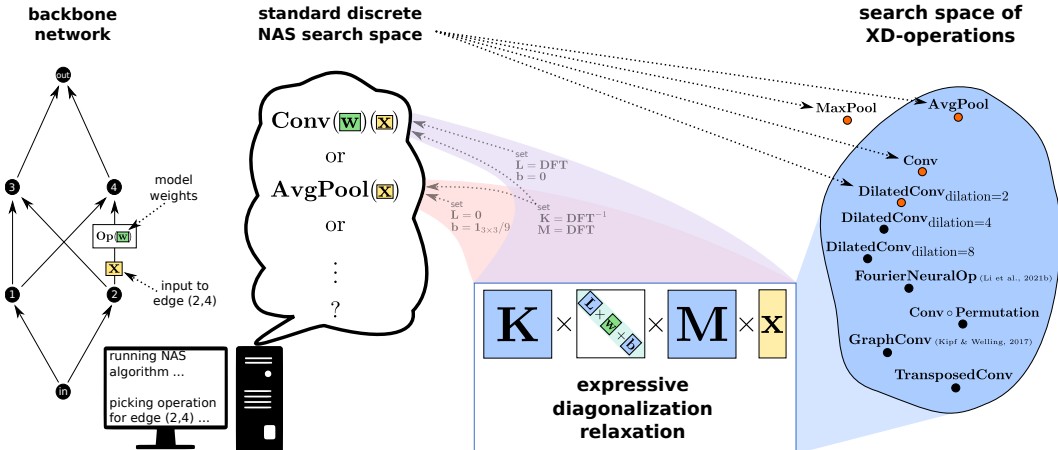

Figure 1: Diagram of our search space depicting a NAS method picking an operation for an edge in a backbone network (left). Instead of choosing from a discrete search space, we use a relaxation based on the convolution's diagonalization by the discrete Fourier transform in which the DFTs are replaced by K-matrices [10] **K**, **L**, and **M** (middle); these are the main architecture parameters of our new search space over Expressive Diagonalization (XD) operations. This space contains most operations considered in standard NAS and many other important operations in a variety of domains (right).

*K-matrices* [10] yields the set of **Expressive Diagonalization (XD) Operations**, which comprise a large search space containing various types of grid-based convolutions and pooling, permutations, transposed convolutions, certain kinds of graph convolutions, the Fourier Neural Operator (FNO) [30], and infinitely many more. This broad expressivity reflects the key insight of our work: that many of the most important neural operations in ML consist of multiple channels that apply weights **w** to inputs **x** by computing

$$\mathbf{K} \operatorname{diag}(\mathbf{L}\mathbf{w})\mathbf{M}\mathbf{x} \tag{1}$$

where the matrices **K**, **L**, and **M** are *efficient* (to represent and apply) and shared across channels.

We leverage XD-operations to take critical steps towards a broader NAS that enables the discovery of good design patterns with limited human specification from data in under-explored domains. To do so we develop a simple procedure which transforms any backbone convolutional neural network (CNN) into an architecture search space by replacing its operations with XD-operations. This space is then searched using a simple weight-sharing algorithm that needs only a small amount of tuning to find effective operations. As a simple first demonstration, we show that XD-operations yield models that are 15% more accurate than standard discrete search spaces on *permuted* CIFAR-10, highlighting the fragility of standard NAS operation spaces on new datasets, and thus the need for XD-operations.

As our main evaluation, we demonstrate the effectiveness of XD-operations in a series of applications showing that, starting from vanilla CNNs, they consistently outperform custom-designed operations.

- **Learning to solve partial differential equations (PDEs):** when substituted into a simple CNN backbone, XD-operations outperform convolutions and the dense prediction NAS method Auto-DeepLab [31], and even achieve lower error than custom-designed, state-of-the-art operations (FNOs [30]) across three problems with different dimensionalities (Burgers' equation, Darcy Flow, and Navier-Stokes). Our method also maintains consistent performance across different resolutions, a major stated advantage of FNOs over previous methods.

- **Protein folding:** on the task of predicting residue distances in a polypeptide chain—a key component of the protein folding problem—we substitute XD-operations into vanilla ResNets and achieve lower error than cyclically-dilated ResNets adapted specifically for this setting [1]. Furthermore, our ResNet-34 XD outperforms the reported error of the much deeper Dilated ResNet-258.

- **Music modeling:** on two next-note prediction tasks, we show that substituting XD-operations into an undilated CNN outperforms temporal convolutional networks (TCNs)—exponentially-dilated 1d CNNs that themselves outperform standard convolutional and recurrent networks [5].

Code to reproduce these results is available here: `https://github.com/nick11roberts/XD`. Software to apply XD-operations can be found here: `https://github.com/mkhodak/relax`.

**Related Work**  AutoML is a well-studied area, with most work focusing on fairly small hyperparameter spaces [7, 24] or on NAS [12]. Most NAS operation spaces only contain a few operations such as convolutions [32, 33, 49, 11], which may not be useful for domains where CNNs are ineffective. Applications of NAS outside vision largely follow the same pattern of combining human-designed operations [35, 43]. On the other extreme, AutoML-Zero [37] demonstrates the possibility of evolving all aspects of ML from scratch. We seek to establish a middle ground with large and domain-agnostic search spaces that still allow the use of well-tested methods, e.g. stochastic gradient descent (SGD).

Several papers have generalized the DFT to replace layers in deep nets [9, 3, 2, 10] in order to speed up or add structure to models while *reducing* expressivity. In contrast, we can replace *convolutions* and other layers while *increasing* expressivity by extending their diagonalization via K-matrices. As discussed in Section 2, using K-matrices for this directly is inefficient for input dimension $> 1$.

## 2  The Expressive Diagonalization Relaxation

In this section we overview our main contribution: a large, general search space of neural operations. Formally, we view an architecture as a *parameterizable* object—a mapping from model weights to functions—described by a *labeled* directed acyclic graph (DAG) $\mathcal{G}(V, E)$. Each edge in $E$ has the form $(u, v, \mathbf{Op})$, where $u, v \in V$ are nodes and $\mathbf{Op}$ is an operation that can be parameterized to define some transformation of the representation at node $u$; node $v$ aggregates the outputs of its incoming edges into a new representation. For example, the popular ResNet architecture [15] has many nodes with two incoming edges, one labeled by the convolution operation $\mathbf{Conv}$ and one by the identity (skip-connect) $\mathbf{Id}$, whose outputs it sums and passes to outgoing edges with the same labels. Each architecture has a source node taking in input data and an output node returning a prediction.

Neural architecture search is the problem of automatically selecting an operation for each edge of $\mathcal{G}$ to optimize an objective.[1] For each edge $e \in E$ a NAS algorithm must pick one element of a *search space* $\mathcal{S} = \{\mathbf{Op}_a \,|\, a \in \mathcal{A}\}$ of operations specified by architecture parameters $a \in \mathcal{A}$ to assign to $e$; in past work, $\mathcal{A}$ usually indexes a small set of operations. As an example, we will refer to a variant[2] $\mathcal{S}_{\mathbf{discrete}}$ of the DARTS search space with parameters $\mathcal{A}_{\mathbf{discrete}} = \{1, \ldots, 8\}$ where each operation is one of $\mathbf{Zero}, \mathbf{Id}, \mathbf{MaxPool}_{3\times3}, \mathbf{AvgPool}_{3\times3}, \mathbf{Conv}_{3\times3 \text{ or } 5\times5}$, or $\mathbf{DilatedConv}_{3\times3,2 \text{ or } 5\times5,2}$ [32].

Our main contribution is a novel family of operations that comprise a search space containing almost all these operations, in addition to many others that have been found useful on different types of data. The starting point of our construction of these XD-operations is the simple observation that all the operations $\mathbf{Op} \in \mathcal{S}_{\mathbf{discrete}}$ listed above except $\mathbf{MaxPool}_{3\times3}$ are *linear*, i.e. for any model weights $\mathbf{w}$ there exists a matrix $\mathbf{A_w}$ such that for all inputs $\mathbf{x}$ we have $\mathbf{Op}(\mathbf{w})(\mathbf{x}) = \mathbf{A_w}\mathbf{x}$. More specifically, all seven of them return convolutions: to see this note that $\mathbf{Zero}, \mathbf{Id}$, and $\mathbf{AvgPool}_{3\times3}$ each apply a convolution with filter $\mathbf{0}_{1\times1}, \mathbf{1}_{1\times1}$, and $\mathbf{1}_{3\times3}/9$, respectively. This means that most of the operations in the DARTS search space—which is representative of NAS operation spaces in computer vision—share the convolution's diagonalization by the discrete Fourier transform (DFT). Formally, if $\mathbf{A_w} \in \mathbb{R}^{n^2 \times n^2}$ is the matrix representing a 2d convolution with filter $\mathbf{w} \in \mathbb{R}^{\mathbf{k}}$ of kernel size $\mathbf{k} \in [n]^2$, then for any 2d input $\mathbf{x} \in \mathbb{R}^{n^2}$ we have

$$\mathbf{Conv}(\mathbf{w})(\mathbf{x}) = \mathbf{A_w}\mathbf{x} = \mathbf{F}^{-1} \operatorname{diag}(\mathbf{F}\underline{\mathbf{w}}) \mathbf{F}\mathbf{x} \tag{2}$$

Here $[n] = \{1, \ldots, n\}$, $\operatorname{diag}(\mathbf{z})$ denotes the diagonal matrix with entries $\mathbf{z}$, $\underline{\mathbf{w}} \in \mathbb{R}^{n^2}$ is an appropriate zero-padding of $\mathbf{w} \in \mathbb{R}^{\mathbf{k}}$, and $\mathbf{F} \in \mathbb{C}^{n^2 \times n^2}$ is the 2d DFT (a Kronecker product of two 1d DFTs).

This diagonalization explicates both the computational and representational efficiency of the DARTS operations, as the DFT and its inverse can be applied in time $\mathcal{O}(n \log n)$ and stored with $\mathcal{O}(n \log n)$ bits. It also suggests a natural way to dramatically expand the operation space while preserving these efficiencies: just replace matrices $\mathbf{F}$ and $\mathbf{F}^{-1}$ in (2) by any one of a general family of efficient matrices. Doing so yields the single-channel version of our *expressive diagonalization* (XD) operations:

$$\mathbf{XD}^{\mathbf{1}}_\alpha(\mathbf{w})(\mathbf{x}) = \operatorname{Real}(\mathbf{K} \operatorname{diag}(\mathbf{L}\underline{\mathbf{w}}) \mathbf{M}\mathbf{x}) \tag{3}$$

Here architecture parameter $\alpha = (\mathbf{K}, \mathbf{L}, \mathbf{M})$ sets the matrices replacing $\mathbf{F}$ and $\mathbf{F}^{-1}$ in Equation 2.

---

[1]It is often defined as selecting both operations and a graph topology [50], but if the set of operations contains the zero-operation $\mathbf{Zero}$ then the former subsumes the latter.

[2]For memory-efficiency, all convolutions in the original DARTS search space are separable [32].

The main remaining question is the family of efficient matrices to use, i.e. the domain of the architecture parameters $\mathbf{K}$, $\mathbf{L}$, and $\mathbf{M}$. For this we turn to the Kaleidoscope matrices, or *K-matrices* [10], which generalize $\mathbf{F}$ and $\mathbf{F}^{-1}$ to include all computationally efficient linear transforms with short description length, including important examples such as sparse matrices and permutations. To obtain this general family, K-matrices allow the DFT's butterfly factors—matrices whose products yield its efficient implementation—to take on different values. While a detailed construction of K-matrices can be found in the original paper, we need only the following useful properties: they are as (asymptotically) efficient to apply as DFTs, are differentiable and can thus be updated using gradient-based methods, and can be composed (made "deeper") to make more expressive K-matrices.

Specifying that $\mathbf{K}$, $\mathbf{L}$, and $\mathbf{M}$ in Equation 3 are K-matrices largely completes our core contribution: a new search space $\mathcal{S}_{\mathbf{XD}}$ of XD-operations with K-matrix architecture parameters. We give a full multi-channel formalization in $N$ dimensions, as well as an overview of its expressivity, in Section 3. First, we note some key aspects of this new search space:

- **Complexity:** $\mathbf{XD}_\alpha^1(\mathbf{w})$ requires three K-matrices and $\mathcal{O}(1)$ filter weights to represent, i.e. description length $\mathcal{O}(n \log n)$; this is larger than a regular convolution (which has no architecture parameters) but is not quadratic in the input size like a linear layer. Applying $\mathbf{XD}_\alpha^1$ requires multiplication by three K-matrices, yielding a theoretical per-channel time complexity of $\mathcal{O}(n \log n)$, matching the efficiency of convolutions. However, as XD-operations strictly generalize convolutions they are more expensive to apply in-practice; we detail these costs both in the application sections and as appendix table, and we view improving upon them as an important future direction.
- **Initialization:** a crucial advantage of XD-operations is that we can initialize or *warm-start* search using operations with known constructions. In particular, since we can recover convolutions (2) by setting architecture parameters $\mathbf{K} = \mathbf{F}^{-1}$, $\mathbf{L} = \mathbf{F}$, and $\mathbf{M} = \mathbf{F}$ in Equation 3, we can always start search with any CNN backbone. We use this extensively in experiments.
- **K-matrices:** as they contain all efficient linear transforms, K-matrices can represent all functions returned by XD-operations, including convolutions. However, for input dimension and filter size $> 1$ the only known way is to apply K-matrices directly to flattened inputs $\mathbf{x} \in \mathbb{R}^{n^N}$, yielding much worse description length $\mathcal{O}(n^N \log n)$. In contrast, as detailed in Section 3, our diagonalization approach uses Kronecker products to apply DFTs to each dimension separately, yielding description length $\mathcal{O}(n \log n)$. It is thus the first (and in some sense, "right") method to use such matrices to replace convolutions. Furthermore, diagonalization allows us to separate model weights $\mathbf{w}$ from architecture parameters $\alpha$, letting the former vary across channels while fixing the latter.

Finally, we address the fact that the architecture parameters of $\mathcal{S}_{\mathbf{XD}}$ are continuous, not discrete, contrasting with much of the NAS literature. This can be viewed as a natural extension of the weight-sharing paradigm [36], in which continuous relaxation enables updating architecture parameters with gradient methods. For example, many algorithms traverse the relaxed DARTS search space $\tilde{\mathcal{S}}_{\mathbf{discrete}} = \left\{ \sum_{i=1}^8 \lambda_i \mathbf{Op}_i \,|\, \lambda_i \geq 0, \sum_{i=1}^8 \lambda_i = 1 \right\}$, defined via DARTS operations $\mathbf{Op}_i \in \mathcal{S}_{\mathbf{discrete}}$ and architecture parameters $\lambda_i$ in the 8-simplex; most search spaces then require discretizing after search via a rounding procedure that maps from the simplex to $\mathcal{A}_{\mathbf{discrete}}$. Note that the fully continuous nature of XD-operations means that we will only evaluate the final network returned by search. In particular, while some weight-sharing papers also report the correlation between true architecture performance and that indicated by the shared weights [46], there is no obvious way to define a ranking or sampling distribution over XD-operations in order to do so. This also means that our final architecture will not be more efficient than the supernet, unlike other weight-sharing methods that do discretize.

## 3 XD-Operations and Their Expressivity

Here we formalize XD-operations and show what operations they include. We first define operations:

**Definition 3.1.** *A* **parameterizable operation** *is a mapping* $\mathbf{Op} : \mathcal{W} \mapsto \mathcal{F}$ *from parameter space* $\mathcal{W}$ *to a space* $\mathcal{F} = \{\mathbf{Op}(\mathbf{w}) : \mathcal{X} \mapsto \mathcal{Y} | \mathbf{w} \in \mathcal{W}\}$ *of* **parameterized functions** *from input space* $\mathcal{X}$ *to output space* $\mathcal{Y}$. *A* **search space** *is a set of operations with the same* $\mathcal{W}$, $\mathcal{X}$, *and* $\mathcal{Y}$.

For example, if $\mathcal{X} = \mathcal{Y} = \mathbb{R}^n$ and $\mathcal{W} = \mathbb{R}^{n \times n}$ then each $\mathbf{W} \in \mathcal{W}$ defines a parameterized linear layer that for each $\mathbf{x} \in \mathcal{X}$ returns $\mathbf{Lin}(\mathbf{W})(\mathbf{x}) = \mathbf{Wx}$. Here $\mathbf{Lin}$ is the parameterizable operation and for each $\mathbf{W}$ the linear map $\mathbf{Lin}(\mathbf{W})$ is the parameterized function.

From Definition 3.1, we say a search space can *express* a specific operation if it contains it. Crucially, the ability of a parameterizable operation $\mathbf{Op}_1$ to express a parameterized function $\mathbf{Op}_2(\mathbf{w})$ output from another operation $\mathbf{Op}_2$ given the right set of weights $\mathbf{w}$ does *not* imply that a search space containing $\mathbf{Op}_1$ can express $\mathbf{Op}_2$. For example, $\mathbf{Lin}(\mathbf{I}_n) = \mathbf{Id}(\mathbf{W}) \ \forall \ \mathbf{W} \in \mathbb{R}^{n \times n}$ but $\mathbf{Lin}(\mathbf{W}) \neq \mathbf{Id}(\mathbf{W}) \ \forall \ \mathbf{W} \neq \mathbf{I}_n$, so a search space containing the linear operation $\mathbf{Lin}$ cannot express the skip-connection $\mathbf{Id}$, despite the fact that $\mathbf{Lin}$ can be parameterized to compute the identity.

**Formalizing Multi-Channel XD-Operations**   Recall the single-channel XD-operation $\mathbf{XD}_\alpha^1$ in Equation 3 specified by three-matrix architecture parameter $\alpha = (\mathbf{K}, \mathbf{L}, \mathbf{M})$. For input dimension $N \geq 1$, every matrix $\mathbf{B} \in \alpha$ is a Kronecker product of $N$ K-matrices of depth $\mathbf{d} \in \mathbb{Z}_+^3$, i.e. $\mathbf{B} = \bigotimes_{i=1}^{N} \mathbf{B}_i$ for K-matrices $\mathbf{B}_i \in \mathbb{C}^{n \times n}$ of depth $\mathbf{d}_{[1]}$, $\mathbf{d}_{[2]}$, or $\mathbf{d}_{[3]}$ for $\mathbf{B} = \mathbf{K}$, $\mathbf{L}$, or $\mathbf{M}$, respectively.[3] Roughly speaking, $\mathbf{XD}_\alpha^1$ can return any linear operation that is diagonalized by K-matrices and is thus efficient to compute and represent, e.g. any convolution (recall we recover the diagonalization of $\mathbf{Conv}(\mathbf{w})$ in Equation 2 by setting $\mathbf{K}$, $\mathbf{L}$, and $\mathbf{M}$ appropriately in Equation 3). However, $\mathbf{XD}_\alpha^1$ cannot represent efficient *parameter-free* operations such as skip-connections and average-pooling, both common in NAS. In particular, the only way to always ignore the model weights $\mathbf{w}$ is to set one of the K-matrices to zero, producing the zero-operation. We avoid this by adding a bias $\mathbf{b} \in \mathbb{C}^{n^N}$ as an architecture parameter, yielding the *biased* single-channel XD-operation:[4]

$$\mathbf{XD}_{\alpha, \mathbf{b}}^1(\mathbf{w})(\mathbf{x}) = \mathrm{Real}\left(\mathbf{K}\,\mathrm{diag}(\mathbf{L}\underline{\mathbf{w}} + \mathbf{b})\mathbf{M}\underline{\mathbf{x}}\right) \tag{4}$$

This lets us define skip-connections (set $\mathbf{K} = \mathbf{M} = \mathbf{I}_{n^N}$, $\mathbf{L} = \mathbf{0}_{n^N \times n^N}$, and $\mathbf{b} = \mathbf{1}_{n^N}$) and average-pooling (set $\mathbf{K} = \mathbf{F}^{-1}$, $\mathbf{L} = \mathbf{0}_{n^N \times n^N}$, $\mathbf{M} = \mathbf{F}$, and $\mathbf{b}$ to be $\mathbf{F}$ multiplied by a pooling filter).

Lastly, we use $\mathbf{XD}_{\alpha, \mathbf{b}}^1$ to construct multi-channel "layers" that pass multiple input features through multiple channels and re-combine them as multiple output features. This follows the primary way of using convolutions in deep nets. The key insight here is that we will share the same parameterizable operation (specified by $\alpha$ and $\mathbf{b}$) across all channels, just as in convolutional layers.

**Definition 3.2.** *Let $a = (\alpha, \mathbf{b}, \mathbf{C})$ be an architecture parameter containing a triple $\alpha = (\mathbf{K}, \mathbf{L}, \mathbf{M})$ of Kronecker products of $N$ K-matrices with depths $\mathbf{d} \in \mathbb{Z}_+^3$, a bias $\mathbf{b} \in \mathbb{C}^{n^N}$, and channel gates $\mathbf{C} \in \mathbb{C}^{c \times c}$.[5] Using "$\bigoplus$" to denote concatenation, the **XD-operation** $\mathbf{XD}_a$ of depth $\mathbf{d}$ specified by $a$ is a parameterizable operation on parameter space $\mathcal{W} = \mathbb{R}^{c \times c \times \mathbf{k}}$ consisting of $c^2$ filters of size $\mathbf{k} \in [n]^N$ that outputs parameterized functions on $\mathcal{X} = \mathbb{R}^{c \times m^N}$ for $m \leq n$ mapping every $\mathbf{x} \in \mathcal{X}$ to*

$$\mathbf{XD}_a(\mathbf{w})(\mathbf{x}) = \bigoplus_{i=1}^{c} \sum_{j=1}^{c} \mathbf{C}_{[i,j]} \mathbf{XD}_{\alpha, \mathbf{b}}^1(\mathbf{w}_{[i,j]})(\mathbf{x}_{[j]}) \tag{5}$$

The last architecture parameter $\mathbf{C}$ allows interpolation between all-to-all layers ($\mathbf{C} = \mathbf{1}_{c \times c}$), e.g. multi-channel convolutions, and layers where each channel is connected to one other channel ($\mathbf{C} = \mathbf{I}_c$), e.g. skip-connections and average-pooling. We note that we use $\mathcal{S}_{\mathbf{XD}}$ to describe the set of operations covered by Definition 3.2 and conclude our construction by discussing two properties:

- **Kernel size:** the weight-space available to an XD-operation is $\mathbb{R}^{c \times c \times n^N}$; however, since we will initialize search with existing CNNs, we will zero-pad to have the same weight-space $\mathbb{R}^{c \times c \times k^N}$ as the convolutions with filter size $k \leq n$ that they replace. This preserves the weight count but also means that if the backbone has $3 \times 3$ filters our search space will *not* contain $5 \times 5$ convolutions. Experimentally, we find that relaxing the constraint to allow this does not significantly affect results on image tasks, so we do not do so in subsequent applications to avoid increasing the weight count.
- **Depth:** an XD-operation's depth is a triple describing the depths of its K-matrices $\mathbf{K}$, $\mathbf{L}$, and $\mathbf{M}$. Increasing it trades off efficiency for expressivity; for example, in the next section we describe operations that we can show are contained in $\mathcal{S}_{\mathbf{XD}}$ if $\mathbf{L}$ or $\mathbf{M}$ have depth $> 1$. By default we will set the depth to be the minimum needed to initialize search with the backbone operation.

---

[3] A depth-$d$ K-matrix is a product of $d$ depth-1 K-matrices.

[4] Zero-padding $\mathbf{x}$ as well lets the input to be smaller than the output if needed, e.g. for transposed convolutions.

[5] For simplicity we formalize the case where all $N$ dimensions have the same input size and there is an identical number $c$ of input and output channels; both are straightforward to extend.

**Expressivity of XD-Operations** For many papers that replace deep net layers with efficient linear transforms [34, 10], the question of expressivity comes down to the transform capacity. For example, layers with a K-matrix in every channel can represent a different transform in each, thus allowing the output to be any combination of efficient linear operations. Our case is less straightforward since we care about expressivity of the search space, not of parameterized functions, and our approach is less-expressive *by design* as all channels share K-matrices $\mathbf{K}$, $\mathbf{L}$, and $\mathbf{M}$. The latter can be thought of as a useful inductive bias on NAS: the set of XD-operations is still much broader than the set of convolutions, but the way in which model weights are applied is the same across all channels.

Expressivity results are a way to see if this bias is useful or constraining. Here we summarize some important operations that are 1d XD-operations; proofs can be found in the appendix and are straightforward to extend to multi-dimensional inputs. Formally, there exists $\mathbf{d} \in \mathbb{Z}_+^3$ such that the set of XD-operations of depth $\mathbf{d}$ over weights $\mathcal{W} = \mathbb{R}^{c \times c \times k}$ and inputs $\mathcal{X} = \mathbb{R}^m$ for $m \leq n$ contains

1. convolutions with filter size $\leq k$, dilation $\leq \lfloor \frac{n-1}{k-1} \rfloor$, stride $\leq n - 1$, and arbitrary channel groups.
2. parameter-free operations $\mathbf{Id}$, $\mathbf{Zero}$, and $\mathbf{AvgPool}_s$ for any kernel size $s \leq n$.
3. composing 1 or 2 with multiplication of all input or output channels by a bounded-depth K-matrix.

Note this does not account for *all* important XD-operations, e.g. we show in the appendix that they also express Fourier Neural Operators [30] with $\leq \lfloor k/2 \rfloor$ modes and any transposed convolutions whose stride equals the dilated kernel size.[6] Still, the first two items account for non-separable variants of most operations considered in past NAS work in computer vision, excluding the nonlinear $\mathbf{MaxPool}$ [47, 11]. Note depthwise-separable convolutions *are* contained in the set of compositions of XD-operations. The third item implies that XD-operations can express the basic and diffusion graph convolutions over fixed graphs [21, 27]: both are point-wise convolutions composed with sparse multiplication by a modified adjacency matrix, which K-matrices can represent efficiently.

As a concrete example, consider dilated convolutions, which for $k > 1$ and dilation factor $d \geq 1$ apply filters of effective size $(k-1)d + 1$ with nonzero entries separated by $d - 1$ zeros. One could hope to express the application of $\mathbf{DilatedConv}_{k,d}$ to an input $\mathbf{x} \in \mathbb{R}^n$ in the single-channel setting as $\mathbf{F}^{-1} \operatorname{diag}(\mathbf{F} \operatorname{diag}(\mathbf{p}_{k,d}) \underline{\mathbf{w}}) \mathbf{F} \mathbf{x}$, where $\mathbf{p}_{k,d} \in \{0,1\}^n$ zeroes out appropriate entries of $\underline{\mathbf{w}}$, but this requires filter size $(k-1)d + 1 > k$, increasing the number of weights. Instead, we can use a permutation $\mathbf{P}_{k,d} \in \{0,1\}^{n \times n}$ before the DFT to place the $k$ entries of $\underline{\mathbf{w}}$ into dilated positions:

$$\mathbf{DilatedConv}_{k,d}(\mathbf{w})(\mathbf{x}) = \mathbf{F}^{-1} \operatorname{diag}(\mathbf{F} \mathbf{P}_{k,d} \underline{\mathbf{w}}) \mathbf{F} \mathbf{x} \qquad (6)$$

As permutations are depth-2 K-matrices [10], we can express $\mathbf{DilatedConv}_{k,d}$ with an XD-operation of depth $(1, 3, 1)$, with $\mathbf{K} = \mathbf{F}^{-1}$, $\mathbf{L} = \mathbf{F} \mathbf{P}_{k,d}$, and $\mathbf{M} = \mathbf{F}$.

## 4 Finding and Evaluating XD-Operations

This section outlines a simple procedure that we use to evaluate XD-operations. Recall that NAS methods specify architectures by assigning operations to each edge $(u, v, \mathbf{Op})$ of a computational graph. We aim to simultaneously find good operations and model weights, a goal distinct from the classic *two-stage* NAS formulation, which finds assignments in an initial search phase before training the resulting architecture from scratch [47]. However, the use of weight-sharing [36] extends NAS to *one-shot* objectives where weights and architectures are jointly optimized. Under weight-sharing, architecture parameters become weights in a larger "supernet," extending the hypothesis class [25].

To assess XD-operations directly we assume the user provides a starter network with existing edge labels $\mathbf{Op}_{u,v}$ as a backbone. We transform this into a weight-sharing supernet by reparameterizing each operation $\mathbf{Op}_{u,v}$ as an XD-operation $\mathbf{XD}_{a_{u,v}}$ with architecture parameter $a_{u,v}$. Then we simultaneously train both $a_{u,v}$ and the model weights $\mathbf{w}_{u,v}$ associated with each edge as follows:

- **Architecture parameters** $a_{u,v}$ are initialized using the original operation used by the CNN backbone by setting $\mathbf{Op}_{u,v} = \mathbf{XD}_{a_{u,v}}$; $a_{u,v}$ is then updated via SGD or Adam [20]. We tune step-size, momentum, and the number of "warmup" epochs: initial epochs during which only model weights $\mathbf{w}_{u,v}$ are updated. This can be viewed as a specialized step-size schedule.
- **Model weights** $\mathbf{w}_{u,v}$ are initialized and updated using the routine provided with the backbone.

---

[6]This restriction still includes transposed convolutions used in well-known architectures such as U-Net [38].

Table 1: Search space comparison on CIFAR-10. Validation accuracies are averages of three trials. While we use small CNNs for exploration, XD-operations can also be used with high-performance backbones to obtain $> 95\%$ accuracy (c.f. the appendix).

| Backbone search space | CIFAR-10 | Permuted CIFAR-10[*] | Cost (hours[†]) |
|---|---|---|---|
| **LeNet** | $75.5 \pm 0.1$ | $43.7 \pm 0.5$ | 0.3 |
| $\tilde{\mathcal{S}}_{\mathbf{discrete}}$ | $75.6 \pm 3.4$ | $47.7 \pm 1.0$ | 1.0 |
| $\mathcal{S}_{\mathbf{XD}}$ | $77.7 \pm 0.7$ | $63.0 \pm 1.0$ | 0.9 |
| **ResNet-20** | $91.7 \pm 0.2$ | $58.6 \pm 0.7$ | 0.6 |
| $\tilde{\mathcal{S}}_{\mathbf{discrete}}$ | $92.7 \pm 0.2$ | $58.0 \pm 1.0$ | 5.3 |
| $\mathcal{S}_{\mathbf{XD}}$ | $92.4 \pm 0.2$ | $73.5 \pm 1.6$ | 5.6 |

[*] No data augmentation used in the permuted case.

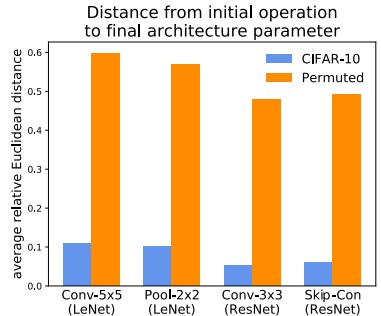

Distance from initial operation to final architecture parameter

Figure 2: On permuted images, where convolutions are not the "right" operation, we find XD-operations that are farther away from the operations of the initial CNN backbone.

This approach allows us to use established topologies and optimizers while searching for new operations, thus aligning with the goal for Sections 5, 6, and 7: to improve upon the CNN backbones that practitioners often use as a first attempt. As a simple example, we start by applying the procedure to image classification. Since this is not the main objective of our work, we treat it as a warmup and consider two datasets: CIFAR-10 and a variant where the images' rows and columns are permuted. On CIFAR-10 we do *not* expect to see much improvement from XD-operations over the CNN backbone used to initialize search, as convolutions are already the "right" operation for images. On the other hand, the "right" operation on permuted data, at least in layer one, is an inverse permutation followed by convolution; as this is an XD-operation[7], here we do hope to see improvement.

Using LeNet [23] and ResNet-20 [15] as backbones, we compare applying our algorithm to XD-operations with two baselines: (1) using just the backbone CNN and (2) applying a similar method to the relaxed set $\tilde{\mathcal{S}}_{\mathbf{discrete}}$ of DARTS operations from Section 2. To optimize over $\tilde{\mathcal{S}}_{\mathbf{discrete}}$ we take an approach similar to DARTS: parameterize the simplex using a softmax and apply Adam. We experiment with both a uniform initialization and one biased towards the backbone's operation. While both $\mathcal{S}_{\mathbf{XD}}$ and $\mathcal{S}_{\mathbf{discrete}}$ contain LeNet's $\mathbf{Conv}_{5\times5}$ and ResNet's $\mathbf{Conv}_{3\times3}$ and $\mathbf{Id}$, for LeNet's $\mathbf{MaxPool}_{3\times3}$ layer we initialize with the closest operation. For direct comparison, both search spaces employ weights with maximum filter size $5 \times 5$ and for both we evaluate the shared weights rather than retraining, which we find hurts $\tilde{\mathcal{S}}_{\mathbf{discrete}}$. We set the XD-operations' depth to $\mathbf{d} = \mathbf{3}_3$ to express the dilated convolutions in $\mathcal{S}_{\mathbf{discrete}}$ and convolutions composed with permutations.

In Table 1, we see that while both the relaxed discrete NAS operations and XD-operations perform comparably on regular images, XD-operations achieve around 15% better accuracy with both backbones when the images are permuted.[8] Note that even networks obtained by running state-of-the-art NAS procedures such as GAEA PC-DARTS [25] and DenseNAS [13] on permuted CIFAR-10 achieve only 66.3% and 61.6% accuracy, respectively, despite using millions more parameters than ResNet-20. While it is not straightforward to understand the recovered XD-operations that perform so well, we can use the relative Euclidean distance of their architecture parameters from initialization as a proxy for novelty; in Figure 2 we see that on regular images our procedure finds operations that are quite similar to convolutions, but on permuted data they are much further away. These results show that to enable NAS on diverse data, we will need a search space that contains truly novel operations, not just combinations of existing ones. In the remainder of the paper, we study more diverse and realistic tasks that show further evidence that $\mathcal{S}_{\mathbf{XD}}$ is a strong candidate for this.

---

[7]Recall $\mathcal{S}_{\mathbf{XD}}$ includes compositions of convolutions with multiplication by a K-matrix, e.g. a permutation.
[8]Full accuracy can be recovered via an auxiliary loss encouraging permutation-like K-matrices [10].

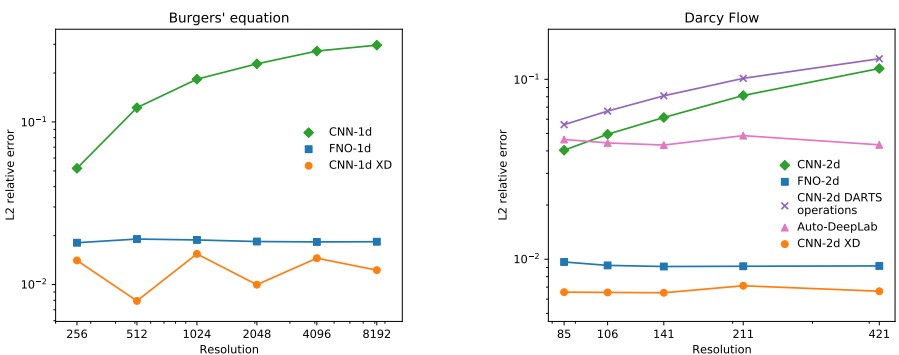

Figure 3: Relative error on Burgers' equation (left) and Darcy Flow (right) across different resolutions.

## 5 Application: Learning to Solve Partial Differential Equations

As our first non-vision application, we consider the task of solving PDEs, an important application area of ML in the natural sciences [28, 29, 41]. In our setup, data generated by classical PDE solvers is used to learn functions from some initial condition or setting to the corresponding PDE solution, with the goal of replacing the solver by a deep net forward pass; the latter can be orders of magnitude faster. A recent state-of-the-art approach for this introduces Fourier Neural Operators [30], operations that significantly improve upon previous neural approaches across three different PDE settings. To evaluate the ability of XD-operations to compete with such custom-designed operations starting from simple CNN backbones, we will investigate the same three PDEs that they study: Burgers' equation, Darcy Flow, and the 2d Navier-Stokes equations, which involve 1d, 2d, and 3d data, respectively. The first two are studied across multiple resolutions, while the last one is studied at different viscosities.

As before, we start with a simple CNN backbone—the type a scientist might use in a first attempt at a solution—and replace all convolutions by XD-operations. We initially hope to do better than this backbone, but ambitiously also hope to compete with the custom-designed FNO. The specific CNN we use is simply the FNO architecture of the appropriate dimension $N$ but with all $N$-dimensional FNOs replaced by $N$-dimensional convolutions; this performs similarly to their CNN baselines [30]. In all cases we compare mainly to the CNN backbone and our reproduction of the FNO results, as the latter exceeds all other neural methods; a complete results table is provided in the appendix. Our reproduction of FNO is slightly worse than their reported numbers for Burgers' equation and slightly better in the other two settings. Note that on the Navier-Stokes equations we only compare to the 3d FNO on the two settings in which we were able to reproduce their approach; moreover, we do *not* compare to their use of a 2d FNO plus a recurrent net in time, but in-principle XD-operations can also be substituted there. In the 2d Darcy Flow case we also include comparisons to DARTS operations in the simple CNN backbone, as in Section 4, and to Auto-DeepLab (AutoDL) [31], a well-known NAS method for dense prediction. For evaluating XD-operations we again follow the procedure in Section 4, in which we tune only the architecture optimizer; notably, we do this only at the lowest resolutions. At all dimensions we use XD-operations of depth $\mathbf{d} = \mathbf{1}_3$; in addition, in dimensions $N > 1$ we fix the architecture biases $\mathbf{b}$ and channel gates $\mathbf{C}$ to $\mathbf{0}$ and $\mathbf{1}$, respectively, to conserve memory at higher resolutions. At lower ones we find that the performance difference is negligible.

We report our results for the Burger's equation and Darcy Flow in Figure 3; for 2d Navier-Stokes the results are in Table 2. In all cases we dramatically outperform the CNN backbone used to initialize XD-operations; furthermore, we also achieve better error than FNO, despite it being custom-made for this problem. In particular, we find that XD-operations have higher *training error* but generalize better (c.f. the appendix). Figure 3 also shows that XD-operations perform consistently well across resolutions, a major advantage of FNOs over previous methods, whose performance was tightly coupled to the discretization [30]. Notably, CNN performance worsens with higher resolution, unlike that of XD and FNO. Finally, we also substantially outperform DARTS operations and AutoDL in 2d, although the latter is at least consistent across resolutions. These results provide strong evidence that XD-operations are a useful search space for discovering neural operations, even in domains where the convolutions used to initialize them perform much worse than state-of-the-art. Note that these results do come at a cost of slower training and inference: XD-operations are roughly an order of magnitude slower than FNOs, despite having fewer parameters in 2d and 3d. This still yields solvers one-to-two orders of magnitude faster than classical solvers, maintaining usefulness for the problem.

Table 2: Relative test error on the 2d Navier-Stokes equations at different settings of the viscosity $\nu$ and time steps $T$. Best results in each setting are **bolded**.

|  | $\nu = 10^{-4}, T = 30$ | $\nu = 10^{-5}, T = 20$ |
|---|---|---|
| CNN-3d (our baseline) | 0.325 | 0.278 |
| FNO-3d (reproduced) | 0.182 | 0.177 |
| **CNN-3d XD** (ours) | **0.172** | **0.168** |

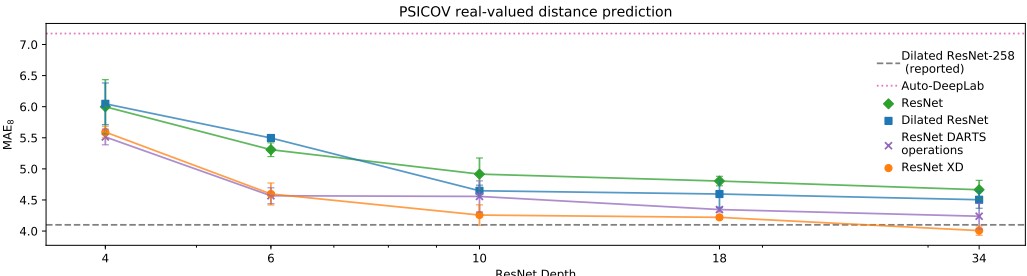

Figure 4: ResNet XD outperforms both baseline and dilated ResNets on PSICOV. At the highest depth we test we also outperform the reported $\mathrm{MAE}_8$ of the much deeper Dilated ResNet-258 [1].

# 6   Application: Real-Valued Distance Prediction for Protein Folding

As a second scientific application, we consider the task of inferring the 3d "folded" structure of a polypeptide chain, which yields important insights into the function of the resulting protein [18]. This problem is a high-priority challenge in biology and has recently seen significant ML-driven advances from deep learning methods such as AlphaFold [40, 19] and PDNET [1]. These typically involve training a network to predict pairwise physical distances between residues in the chain. We work with the PDNET benchmark, which consists of a training set of 3,356 proteins, a validation set of 100 of proteins, and the PSICOV [18] test set of 150 proteins. PDNET is designed to be more accessible than datasets used by large-scale methods such as AlphaFold, which are not always publicly available and/or require massive compute [40, 19]. We follow the PDNET training procedure [1] and evaluate test set performance using their $\mathrm{MAE}_8$ metric for assessing long-range distances.

As before we start with simple CNN backbones—in this case ResNets. We choose this to compare most directly to the custom-designed architecture used by PDNET, consisting of a Dilated ResNet characterized by its use of a cyclically increasing dilation rate across ResNet blocks [1]. At a sufficient depth, the Dilated ResNet is shown to outperform a standard pre-activation ResNet adapted to this task [1]. Our goal will be to see whether we can start with the vanilla ResNet and use XD to outperform both it and the specialized Dilated ResNet. We also aim to outperform the DARTS operations baseline from the previous two sections as well as the AutoDL NAS approach for dense prediction. We use XD-operations of depth $\mathbf{d} = \mathbf{1}_3$ and fix the architecture biases and channel gates as before to conserve memory. We evaluate architectures of different depths—4, 6, 10, 18, and 34—by varying the number of ResNet blocks used in the backbone architecture and baseline.

We report the results as averages across three trials for each depth in Figure 4. Notably, while Dilated ResNet slightly outperforms ResNet, ResNet XD outperforms both dilated and standard ResNets at all depths. This provides further evidence that XD-operations can outperform specialized operations for diverse domains, even when initialized naively as standard convolutions. XD also outperforms AutoDL, which does poorly, and DARTS operations, except at the two smaller depths where performance is similar. Moreover, our ResNet-34 XD's $\mathrm{MAE}_8$ of 4.0 also improves upon PDNET's reported $\mathrm{MAE}_8$ of 4.1 attained by the much deeper Dilated ResNet-258 [1]; however, in our reproduction Dilated ResNet-258 achieved an $\mathrm{MAE}_8$ of 3.5. Given the trend in Figure 4, where XD-operations consistently improve the backbone architecture of the same depth, we conjecture that ResNet-258 XD could further improve upon this result. We leave scaling XD-operations to such deeper networks to future work.

Table 3: XD-operations compared to recent results in music modeling. We report average loss across three trials. The best result on each task is **bolded**.

| Method (source) | JSB Chorales | Nottingham |
|---|---|---|
| Best recurrent [5] | 8.43 | 3.29 |
| TCN [5] | 8.10 | 3.07 |
| Transformer [44] | - | 3.34 |
| R-Transformer [44] | - | **2.37** |
| Undilated TCN (our baseline) | $8.16 \pm 0.04$ | $3.23 \pm 0.02$ |
| TCN (reproduced) | $8.17 \pm 0.01$ | $2.97 \pm 0.01$ |
| **Undilated TCN XD** (ours) | $\mathbf{8.07 \pm 0.01}$ | $2.84 \pm 0.02$ |

## 7  Application: Music Modeling

Our final application is to music modeling, i.e. learning to predict the next note from sheet music [4]. The dominant approaches for such tasks are recurrent nets [16] and Transformers [42], but recent work has shown that specially-designed convolutional models can also be made competitive at similar model sizes [5, 6]. We will consider the temporal convolutional network (TCN) [5], which improves upon a regular CNN by having the dilation factor grow exponentially across layers. The tasks we study are on the JSB Chorales and Nottingham corpora, used in the original evaluation of TCNs [5]. As the baseline we take the TCN and set all dilation factors to one (undilated); our goal will be to start with this undilated network and match or outperform the custom dilation design of the TCN.

The results presented in Table 3 show that we achieve this goal, as we outperform both the undilated baseline and the TCN on both tasks. While the simple undilated backbone that we initialize with turns out to already match the TCN on JSB Chorales, on Nottingham our approach demonstrates that XD-operations can be used to outperform hand-designed architectures starting from vanilla CNNs.[9] Where possible we also compare to other known results; XD-operations outperforms all of these except the R-Transformer [44], a model combining recurrent nets and self-attention, on Nottingham.

Together with our results on PDEs and proteins, our study of music modeling provides further evidence that XD-operations can effectively find good operations using standard backbones on diverse tasks. One notable difficulty here is causality enforcement: making sure the input data does not contain the target when predicting the next entry. While TCNs can efficiently do so via temporal shifts, we do it in a brute-force manner by treating sequences of length $n$ as $n - 1$ data-points with masked targets. This is expensive and thus limits our evaluation to small music tasks. A fruitful direction for future work is thus to examine whether it is possibly to directly enforce causality in XD-operations, e.g. by forcing architecture parameters $\mathbf{K}$ and $\mathbf{M}$ to be lower triangular; since a product of lower triangular matrices is again lower triangular, the entire operation is then a multiplication of the input sequence by a lower triangular matrix, which suffices to prevent causality violations.

## 8  Conclusion

This work aims to transition NAS from combining existing operations designed for vision and text to finding novel and effective operations in many domains. To do so we introduced a new search space of XD-operations and demonstrated its effectiveness on diverse tasks. Combining XD-operations with standard topology-search NAS, warm-starting search from non-standard operations such as graph convolutions and FNOs,[10] improving the computational limitations described earlier, and constructing spaces containing missing operations such as BatchNorm [17] and self-attention [42] are all promising future directions. Finally, note that our goal—lowering the barrier for applying ML—necessarily comes with the possibility of misuse. Mitigating this involves developing tools for application-specific concerns, e.g. privacy and fairness, that go beyond the error metrics we target.

---

[9]In the appendix we report similar improvements on two other tasks on which TCNs were evaluated—permuted MNIST and Penn TreeBank—that we do not discuss in detail as our focus is on under-explored tasks.

[10] In this direction, we found that initializing XD with FNO did *worse* than initializing with convolutions on Burgers' equation and Darcy Flow, a surprising result given how much better FNO is than the baseline CNN. Similarly, initializing XD with convolutions dilated as in the original TCN did not lead to significant improvement, except in one setting, over undilated initialization. See the appendix for more details and results.

## Acknowledgments

We thank Maria-Florina Balcan, Jeremy Cohen, and Tian Li for helpful advice on early versions of this paper and anonymous reviewers for suggested improvements. This work was supported in part by DARPA under cooperative agreements FA875017C0141 and HR0011202000, NSF grants CCF-1535967, CCF-1910321, IIS-1618714, IIS-1705121, IIS-1838017, IIS-1901403, and IIS-2046613, a Microsoft Research Faculty Fellowship, a Bloomberg Data Science research grant, an Amazon Research Award, an AWS Machine Learning Research Award, a Facebook Faculty Research Award, funding from Booz Allen Hamilton Inc., a Block Center Grant, a Carnegie Bosch Institute Research Award, and a Two Sigma Fellowship Award. We also gratefully acknowledge the support of NIH under No. U54EB020405 (Mobilize), NSF under Nos. CCF1763315 (Beyond Sparsity), CCF1563078 (Volume to Velocity), and 1937301 (RTML); ONR under No. N000141712266 (Unifying Weak Supervision); the Moore Foundation, NXP, Xilinx, LETI-CEA, Intel, IBM, Microsoft, NEC, Toshiba, TSMC, ARM, Hitachi, BASF, Accenture, Ericsson, Qualcomm, Analog Devices, the Okawa Foundation, American Family Insurance, Google Cloud, Swiss Re, Total, the HAI-AWS Cloud Credits for Research program, the Stanford Data Science Initiative (SDSI), and members of the Stanford DAWN project: Facebook, Google, and VMWare. The Mobilize Center is a Biomedical Technology Resource Center, funded by the NIH National Institute of Biomedical Imaging and Bioengineering through Grant P41EB027060. The U.S. Government is authorized to reproduce and distribute reprints for Governmental purposes notwithstanding any copyright notation thereon. Any opinions, findings and conclusions, or recommendations expressed in this material are those of the authors and do not necessarily reflect the views of DARPA, NSF, NIH, ONR, or any other funding agency.

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
