# A   Expressivity Results

Here we collect results on the expressivity of the set of **XD**-operations. For simplicity, our results will be in the following single-dimensional ($N = 1$) setting:

**Setting A.1.** *We consider input spaces of form $\mathcal{X} = \mathbb{R}^{c \times m}$ for input size $m \in \mathbb{N}$ and channel count $c \in \mathbb{N}$ and parameter spaces $\mathcal{W} = \mathbb{R}^{c \times c \times k}$ for filter size $k \in [n]$, where output size $n \geq m$ is a power of 2.*

It is straightforward to extend the results to multiple dimensions using Kronecker products and to input sizes other than powers of two using padding. Note that all of our results will also assume a circular padded domain.

## A.1   Convolutions

**Definition A.1.** *A* **convolution** *in Setting A.1 with filter size $k$, dilation $d \in [\lfloor \frac{n-1}{k-1} \rfloor]$, stride $s \in [n-1]$, and channel groups described by a matrix $\mathbf{B} \in \{0, 1\}^{n \times n}$ s.t. $\mathbf{B}_{[i,j]} = 1$ if channels $i$ and $j$ are in the same group and 0 otherwise is a parameterizable operation that for any weight $\mathbf{w} \in \mathcal{W}$ outputs a function mapping every $\mathbf{x} \in \mathcal{X}$ to*

$$\frac{1}{n} \begin{pmatrix} \operatorname{diag}(\mathfrak{a}_s(\underline{\mathbf{1}_{\lceil \frac{n}{s} \rceil}})) \sum_{j=1}^{c} \mathbf{B}_{[1,j]} \mathbf{F}_n^{-1} \operatorname{diag}(\mathbf{F}_n \mathfrak{a}_d(\underline{\mathbf{w}_{[1,j]}})) \mathbf{F}_n \mathbf{x}_{[j]} \\ \vdots \\ \operatorname{diag}(\mathfrak{a}_s(\underline{\mathbf{1}_{\lceil \frac{n}{s} \rceil}})) \sum_{j=1}^{c} \mathbf{B}_{[c,j]} \mathbf{F}_n^{-1} \operatorname{diag}(\mathbf{F}_n \mathfrak{a}_d(\underline{\mathbf{w}_{[c,j]}})) \mathbf{F}_n \mathbf{x}_{[j]} \end{pmatrix} \tag{7}$$

*where $\mathbf{F}_n \in \mathbb{C}^{n \times n}$ is the $n \times n$ DFT and $\mathfrak{a}_d : \mathbb{R}^n \mapsto \mathbb{R}^n$ is an atrous permutation of a vector that is equivalent to multiplication by some permutation matrix $\mathbf{P}_d \in \{0, 1\}^{n \times n}$. We will use $\mathbf{Conv}_k$ to denote the case of $d = 1$, $s = 1$, and $\mathbf{B} = \mathbf{1}_{c \times c}$.*

**Claim A.1.** *All multi-channel convolutions of the form given in Definition A.1 are contained in the search space of XD-operations of depth $(1, 3, 1)$.*

*Proof.* Setting the architecture parameters to be $\mathbf{K} = \operatorname{diag}(\mathfrak{a}_s(\underline{\mathbf{1}_{\lceil \frac{n}{s} \rceil}})) \mathbf{F}_n^{-1}$, $\mathbf{L} = \mathbf{F}_n \mathbf{P}_d$, $\mathbf{M} = \mathbf{F}_n$, $\mathbf{b} = \mathbf{0}_n$, and $\mathbf{C} = \mathbf{B}$, and noting that (a) the DFT and its inverse are both depth 1 K-matrices, (b) multiplying a K-matrix by a diagonal matrix is another K-matrix of the same depth, and (c) permutation matrices are K-matrices of depth 2 yields the result. These three facts can be found in the original paper [10]. $\qquad\square$

**Remark A.1.** *Note that for the case of dilation $d = 1$ the result in Claim A.1 holds with depth $\mathbf{1}_3$.*

## A.2   Parameter-Free Operations

**Definition A.2.** *The* **skip-connection** *in Setting A.1 is parameterizable operation that outputs a function mapping every $\mathbf{x} \in \mathcal{X}$ to itself. The* **zero-operation** *in Setting A.1 is parameterizable operation that outputs a function mapping every $\mathbf{x} \in \mathcal{X}$ to $\mathbf{0}_{c \times n}$.*

**Claim A.2.** *The skip-connection and zero-operation are both contained in the search space of XD-operations of depth $\mathbf{1}_3$.*

*Proof.* For both set the architecture parameters to be $\mathbf{K} = \mathbf{F}_n^{-1}$, $\mathbf{L} = \mathbf{0}_{n \times n}$, $\mathbf{M} = \mathbf{F}_n$, and $\mathbf{C} = \mathbf{I}_c$. To obtain the skip-connection set $\mathbf{b} = \mathbf{1}_n$; to obtain the zero-operation set $\mathbf{b} = \mathbf{0}_n$. $\qquad\square$

**Definition A.3.** *An* **average pooling** *operation in Setting A.1 with filter size $k$, dilation $d \in [\lfloor \frac{n-1}{k-1} \rfloor]$, and stride $s \in [n-1]$ is parameterizable operation outputs a function mapping every $\mathbf{x} \in \mathcal{X}$ to the output of a convolution (as in Definition A.1) with the same filter size, dilation, and stride, channel groups described by $\mathbf{B} = \mathbf{I}_c$, and filters $\mathbf{w}_{[j,j]} = \mathbf{1}_k / k \; \forall \, j \in [c]$.*

**Claim A.3.** *All average pooling operations are contained in the search space of XD-operations of depth $\mathbf{1}_3$.*

*Proof.* Setting the architecture parameters to be $\mathbf{K} = \mathrm{diag}(\mathfrak{a}_s(\mathbf{1}_{\lceil \frac{n}{s} \rceil}))\mathbf{F}_n^{-1}$, $\mathbf{L} = \mathbf{0}_{n \times n}$, $\mathbf{M} = \mathbf{F}_n$, $\mathbf{b} = \mathfrak{a}_d(\mathbf{1}_k/k)$, and $\mathbf{C} = \mathbf{I}_c$ and noting that (a) the DFT and its inverse are both depth 1 K-matrices and (b) multiplying a K-matrix by a diagonal matrix of the same depth is another K-matrix of the same depth yields the result. $\qquad\square$

## A.3 Compositions with Multiplication by a Fixed K-Matrix

**Definition A.4.** *A **fixed linear operation** $\mathbf{Lin_A}$ in Setting A.1 with fixed matrix $\mathbf{A} \in \mathbb{R}^{n \times n}$ is a parameterizable operation that outputs a function mapping every $\mathbf{x} \in \mathcal{X}$ to $\mathbf{Lin_A}(\mathbf{w})(\mathbf{x}) = \begin{pmatrix} \mathbf{Ax}_{[1]} & \cdots & \mathbf{Ax}_{[c]} \end{pmatrix}^T$. For example, $\mathbf{Lin_{I_c}} = \mathbf{Id}$.*

**Definition A.5.** *Let $\mathbf{Op}_1$ and $\mathbf{Op}_2$ be two parameterizable operations in Setting A.1 with $\mathcal{X}$. Then for any weight $\mathbf{w} \in \mathcal{W}$ their **composition** $\mathbf{Op}_1 \circ \mathbf{Op}_2$ outputs the parameterized function $\mathbf{Op}_1(\mathbf{w}) \circ \mathbf{Op}_2(\mathbf{w})$.*

**Claim A.4.** *Let $\mathbf{Op}$ be a parameterizable operation in Setting A.1 that is contained in the set of XD-operations of some depth $\mathbf{d} \in \mathbb{N}^3$ and let $\mathbf{A}$ be a K-matrix of depth $d'$. Then $\mathbf{Op} \circ \mathbf{Lin_A}$ is contained in the set of XD-operations of depth $(\mathbf{d}_{[1]}, \mathbf{d}_{[2]}, \mathbf{d}_{[3]} + d')$ and $\mathbf{Lin_A} \circ \mathbf{Op}$ is contained in the set of XD-operations of depth $(\mathbf{d}_{[1]} + d', \mathbf{d}_{[2]}, \mathbf{d}_{[3]})$.*

*Proof.* Let $\mathbf{K}$ and $\mathbf{M}$ be the first and last K-matrices of the representation of $\mathbf{Op}$ as an XD-operation, which thus have depth at most $\mathbf{d}_{[1]}$ and $\mathbf{d}_{[3]}$, respectively. Then the representation of $\mathbf{Op} \circ \mathbf{Lin_A}$ as an XD-operation is the same except with depth $\mathbf{d}_{[3]} + d'$ K-matrix $\mathbf{MA}$ as the last K-matrix, and similarly the representation of $\mathbf{Lin_A} \circ \mathbf{Op}$ as an XD-operation is the same except with depth $\mathbf{d}_{[1]} + d'$ K-matrix $\mathbf{AK}$ as the first K-matrix. $\qquad\square$

## A.4 Other Named Operations

**Definition A.6.** *Suppose we have a fixed $n$-node graph with adjacency matrix $\mathbf{A}$ and degree matrix $\mathbf{D}$, and let $\hat{\mathbf{A}}$ and $\hat{\mathbf{D}}$ be the adjacency and degree matrices, respectively, of the same graph but with added self-loops. Then regular **graph convolution** [21] in Setting A.1 with $k = 1$ is a parameterizable operation that for any weight $\mathbf{W} \in \mathcal{W}$ outputs a function mapping every $\mathbf{x} \in \mathcal{X}$ to $\hat{\mathbf{D}}^{-\frac{1}{2}} \hat{\mathbf{A}} \hat{\mathbf{D}}^{-\frac{1}{2}} \mathbf{x}^T \mathbf{w}$ and the **diffusion graph convolution** [27] in Setting A.1 with $k = 1$ is a parameterizable operation that for any weight $\mathbf{W} \in \mathcal{W}$ outputs a function mapping every $\mathbf{x} \in \mathcal{X}$ to $\mathbf{D}^{-1} \mathbf{A} \mathbf{x}^T \mathbf{w}$.*

**Claim A.5.** *Suppose $\mathbf{A}$ and $\hat{\mathbf{A}}$ can be represented by K-matrices of depth $d$ and $\hat{d}$, respectively. Then the corresponding graph convolution is contained in the search space of XD-operations of depth $(1, 1, \hat{d} + 1)$ and the corresponding diffusion graph convolution in that of depth $(1, 1, d + 1)$.*

*Proof.* For any $\mathbf{G} \in \mathbb{R}^{n \times n}$ we have $\mathbf{Gx}^T \mathbf{w} = \mathbf{Lin_G}(\mathbf{w})(\mathbf{x})\mathbf{w} = \mathbf{Conv}_1(\mathbf{w})(\mathbf{Lin_G}(\mathbf{w})(\mathbf{x})) = (\mathbf{Conv}_1 \circ \mathbf{Lin_G})(\mathbf{w})(\mathbf{x})$. The result follows by Claims A.1 and A.4, the fact that a K-matrix multiplied by a diagonal matrix is another K-matrix of the same depth, and by substituting $\mathbf{G} = \hat{\mathbf{D}}^{-\frac{1}{2}} \hat{\mathbf{A}} \hat{\mathbf{D}}^{-\frac{1}{2}}$ (for graph convolution) or $\mathbf{G} = \mathbf{D}^{-1} \mathbf{A}$ (for diffusion graph convolution). $\qquad\square$

**Remark A.2.** *Note that the above claim is meaningful because adjacency matrices of realistic graphs are usually sparse and sparse matrices can be efficiently represented as K-matrices [10].*

**Definition A.7.** *A **Fourier neural operator** (FNO) [30] in Setting A.1 with even $k$ and thus $k/2$ modes is a parameterizable operation that for any weight $\mathbf{w} \in \mathcal{W}$ outputs a function mapping every $\mathbf{x} \in \mathcal{X}$ to*

$$\begin{pmatrix} \mathrm{Real}\left(\sum_{j=1}^c \mathbf{F}_n^{-1} \mathrm{diag}(\begin{pmatrix}\mathbf{w}_{[1,j,1:k/2]} + i\mathbf{w}_{[1,j,k/2+1:k]} & \mathbf{0}_{n-k/2}\end{pmatrix}^T)\mathbf{F}_n\mathbf{x}_{[j]}\right) \\ \vdots \\ \mathrm{Real}\left(\sum_{j=1}^c \mathbf{F}_n^{-1} \mathrm{diag}(\begin{pmatrix}\mathbf{w}_{[c,j,1:k/2]} + i\mathbf{w}_{[c,j,k/2+1:k]} & \mathbf{0}_{n-k/2}\end{pmatrix}^T)\mathbf{F}_n\mathbf{x}_{[j]}\right) \end{pmatrix} \qquad (8)$$

**Claim A.6.** *The FNO with $k/2$ modes is contained in the search space of XD-operations of depth $(1, 4, 1)$.*

*Proof.* Setting the architecture parameters to be $\mathbf{K} = \mathbf{F}_n^{-1}$, $\mathbf{L} \in \mathbb{C}^{n \times n}$ the $n$-sparse matrix mapping $\underline{\mathbf{w}}$ to $\left( \mathbf{w}_{[1,j,1:k/2]} + i\mathbf{w}_{[1,j,k/2+1:k]} \quad \mathbf{0}_{n-k/2} \right)^T$, $\mathbf{M} = \mathbf{F}_n$, $\mathbf{b} = \mathbf{0}_n$, and $\mathbf{C} = \mathbf{1}_{c \times c}$, and noting that an $n$-sparse matrix is a depth-4 K-matrix [10] yields the result. $\qquad\square$

**Remark A.3.** *If we allow the parameter space in Setting A.1 to be complex then the FNO with all $k$ modes will be contained in the search space of XD-operations of depth $\mathbf{1}_3$.*

**Definition A.8.** *Each channel of* **transposed convolution** *with stride $d(k-1) + 1$, where $k$ is the kernel size and $d$ is the dilation rate, computes a feature map in which each input element is replaced by that element multiplied by the dilated filter of size $d(k-1) + 1$. The multi-channel extension of this over parameter space $\mathcal{W} = \mathbb{R}^{c \times c \times k}$ is similar to that for standard convolutions.*

**Claim A.7.** *All transposed convolutions with stride equal to the dilated kernel size are contained in the search space of XD-operations of depth $(1, 3, 3)$.*

*Proof.* A transposed convolution is equivalent to a regular convolution with the same filter applied to the input after it has been zero-padded and then permuted to separate all entries by $d(k-1)$ zeros. Since permutations are K-matrices of depth 2 the result follows by Claims A.1 and Claim A.4. $\quad\square$

**Definition A.9.** *A* **depthwise-separable convolution** *in Setting A.1 with filter size $k$ but with parameter space $\mathcal{W} = \mathbb{R}^{c \times k} \times \mathbb{R}^{c \times c}$ is a parameterizable operation that for any weight $\mathbf{w} \in \mathcal{W}$ outputs $\mathbf{Conv}_1(\mathbf{w}_{[2]}) \circ \mathbf{Conv}_{k, \mathbf{I}_c}(\mathbf{w}_{[1]})$, where $\mathbf{Conv}_{k, \mathbf{I}_c}$ denotes the convolution in Definition A.1 with $\mathbf{B} = \mathbf{I}_c$.*

**Remark A.4.** *Since both $\mathbf{Conv}_1$ and $\mathbf{Conv}_{k, \mathbf{I}_c}$ are XD-operations, by definition depthwise-separable convolutions are contained in the search space of composed XD-operations, which by Claim A.2 also contains all of the above operations.*

# B   Practical Complexity of XD-Operations

Table 4: Comparison of the computational and memory costs of XD-operations when substituted for convolutions. For simplicity, we consider cases with 2d inputs and where the channel and bias parameters are fixed.

|  | input | kernel | minutes / epoch | | memory (Gb) | | param. $(\times 10^6)$ | |
|---|---|---|---|---|---|---|---|---|
| Task (backbone) | size | size | **Conv** | **XD** | **Conv** | **XD** | **Conv** | **XD** |
| CIFAR-10 (WRN-40-4) | 32 | 3 | 1.4 | 4.3 | 3.73 | 15.6 | 8.96 | 9.08 |
| Darcy Flow (Conv4*) | 85 | 13 | 0.028 | 0.14 | 4.51 | 5.53 | 0.701 | 0.744 |
| PSICOV (ResNet-18) | 128 | 3 | 5.9 | 11 | 1.50 | 10.7 | 0.038 | 0.549 |

* Four-layer convolutional network with parameterized skip (shortcut) connections derived from the FNO network [30] as described in Section 5.

In this section we report a detailed comparison of computational costs of the XD-operation compared to a convolution; this is presented in Table 4. Due to their familiarity, we present results for tasks that have 2d inputs and thus use 2d convolutions in their default backbone. Note that since XD-operations are more general than convolutions, they must by definition be at least as expensive as convolutions in both computation and memory. While in this paper our focus is on absolute performance using learning metrics (e.g. test error), we view finding a good tradeoff between the performance of XD-operations on certain tasks and convolutions, for example by restricting the expressivity of XD-operations, as important directions for future work.

# C  Experimental Details: CIFAR-10 and Permuted CIFAR-10

Table 5: Architecture optimizer settings on CIFAR-10 tasks. Note that the step-size is updated using the same schedule as the backbone.

| search space | backbone | task | optimizer | initial step-size | warmup epochs | perturb |
|---|---|---|---|---|---|---|
| $\tilde{\mathcal{S}}_{\mathbf{discrete}}$ | LeNet | CIFAR-10 | Adam | 1E-1 | 0 | 0.1 |
| | | Permuted | Adam | 1E-1 | 50 | 0.875 |
| | ResNet-20 | CIFAR-10 | Adam | 1E-3 | 0 | 0.1 |
| | | Permuted | Adam | 1E-1 | 0 | 0.875 |
| $\mathcal{S}_{\mathbf{XD}}$ | LeNet | CIFAR-10 | Adam | 1E-4 | 0 | - |
| | | Permuted | Adam | 1E-3 | 0 | - |
| | ResNet-20 | CIFAR-10 | Adam | 1E-4 | 50 | - |
| | | Permuted | Adam | 1E-3 | 0 | - |

For our experiments with image classification backbones we use the standard CIFAR-10 data [22] and a permuted version where all rows and columns are identically permuted. For unpermuted data we use standard data augmentation [15] while for permuted data we do not use any data augmentation. As specified in Section 4, we keep the training routine of the model weights the same and tune only the architecture optimizer, the settings of which are specified in Table 5. Note that for the DARTS operation space we specify a "perturb" parameter that specifies how unbiased the initial architecture parameters are towards the backbone operation; specifically, we initialize architecture parameters so as to assign one minus this quantity as the weight to the backbone operation, so 0.875 means the initialization is uniform (since $|\tilde{\mathcal{S}}_{\mathbf{discrete}}| = 8$) while 0.1 means the backbone operation is assigned 0.9 of the weight.

## C.1  LeNet

The LeNet backbone we consider consists of two $\mathbf{Conv}_{5\times5}$ layers, each followed by $\mathbf{MaxPool}_{2\times2}$, and two fully connected layers. When warm-starting with XD-operations we use $\mathbf{AvgPool}_{2\times2}$ instead of $\mathbf{MaxPool}_{2\times2}$, while when warm-starting with the DARTS operations we use $\mathbf{MaxPool}_{3\times3}$. For the baseline training routine we use 200 epochs of Momentum(0.9), with the first 100 at learning rate 0.01, the next 50 at 0.005, and the last 50 at 0.001.

## C.2  ResNet-20

We use the implementation and training routine provided here: `https://github.com/akamaster/pytorch_resnet_cifar10`. When replacing operations in the backbone we substitute for both the $\mathbf{Conv}_{3\times3}$ operations and the skip-connections $\mathbf{Id}$; some of the latter are downsampled, which XD-operations can handle as strides.

## C.3  WideResNet-40-4

We use the same implementation as for ResNet-20 but adapt the original WRN training routine [48], except with weight-decay set to $10^{-4}$ (as in ResNet-20); on the regular CIFAR-10 tasks this does not seem to affect performance. To conserve computation and memory, we do not tune the architecture optimizer parameters here and simply use the same ones used for ResNet-20; furthermore, we fix the channel and bias parameters of XD-operations and do not allow the kernel size to be larger the $3 \times 3$. Because of these modifications, we only use our evaluation here as a sanity check for large-network performance of XD-operations and do not include it in the main results.

Table 6: Search space comparison on CIFAR-10. Validation accuracies are averages of three trials.

| Backbone | Search Space | CIFAR-10 | Permuted[*] | Cost (hours[†]) |
|---|---|---|---|---|
| LeNet | backbone | $75.5 \pm 0.1$ | $43.7 \pm 0.5$ | 0.3 |
| | $\tilde{\mathcal{S}}_{\mathbf{discrete}}$ | $75.6 \pm 3.4$ | $47.7 \pm 1.0$ | 1.0 |
| | $\mathcal{S}_{\mathbf{XD}}$ | $77.7 \pm 0.7$ | $63.0 \pm 1.0$ | 0.9 |
| ResNet-20 | backbone | $91.7 \pm 0.2$ | $58.6 \pm 0.7$ | 0.6 |
| | $\tilde{\mathcal{S}}_{\mathbf{discrete}}$ | $92.7 \pm 0.2$ | $58.0 \pm 1.0$ | 5.3 |
| | $\mathcal{S}_{\mathbf{XD}}$ | $92.4 \pm 0.2$ | $73.5 \pm 1.6$ | 5.6 |
| WRN-40-4 | backbone | $95.2 \pm 0.1$ | $64.7 \pm 0.9$ | 4.6 |
| | $\tilde{\mathcal{S}}_{\mathbf{discrete}}$ | $95.2 \pm 0.2$ | $61.3 \pm 1.3$ | 19.9 |
| | $\mathcal{S}_{\mathbf{XD}}$ | $95.0 \pm 0.1$ | $72.9 \pm 0.8$ | 14.3 |
| ResNet-18 | DenseNAS | $94.5 \pm 0.3$ | $61.6 \pm 3.3$ | 3.6 |
| Cell | DARTS[‡] | $96.0 \pm 0.2$ | $66.3 \pm 0.5$ | 28.6 |

[*] No data augmentation used in the permuted case.

[†] On a V100 GPU; time for DARTS Cell is training cost only.

[‡] Search using GAEA PC-DARTS [25]; training using "base" routine [46].

## C.4 DARTS Cell Search

To search the full DARTS search space, which is a standard NAS benchmark, we use GAEA PC-DARTS, a recent state-of-the-art method [25], using code made available by the authors here: `https://github.com/liamcli/gaea_release`. On CIFAR-10 we simply use their best reported cell but evaluate it using the "base" routine [46], i.e. without auxiliary losses or additional data augmentation; this is to obtain fair comparison with the other backbone models. Note that the model is still much larger and the training routine much more intensive. On permuted data we follow the standard three-stage pipeline in which we run search four times, train all four found cells and select the best one, and finally train that cell multiple times.

## C.5 DenseNAS Search

We use the DenseNAS search and evaluation code released by the authors here: `https://github.com/JaminFong/DenseNAS`. While the search space is designed for ImageNet [39], we adapt it to CIFAR-10 by taking the DenseNAS-R1 setting and downscale the input sizes to match 32x32 images used.

# D  Experimental Details: Solving PDEs

For our PDE experiments, we use the FNO code and setup [30] provided here: `https://github.com/zongyi-li/fourier_neural_operator`. We use the same training routine and settings as the backbone architecture for each task and only tune the architecture optimizer. We consider the following hyperparameters for the architecture optimizer: Adam vs. SGD (with or without momentum), initial learning rate, and number of warmup epochs. The final hyperparameters for each task can be found in Table 7. Our CNN backbone is analogous to the FNO architecture used for each problem. In particular, the CNN backbone architecture used for each task is simply the FNO architecture where FNO layers of dimension $N$ with $m$ modes are replaced by $N$-dimensional convolutional layers with filters of size $(m + 1)^N$ and circular padding to match the dimensionality of FNO. In Table 8 and Table 9 we present reported [30], reproduced, and our own results on the 1d Burgers' equation and 2d Darcy Flow.

For AutoDL we use the code and setup provided here: `https://github.com/NoamRosenberg/autodeeplab`. We only conduct search on the lowest resolution and use the resulting architecture at higher resolutions. Search was conducted for 40 epochs, as in the original paper, and the search learning rate was tuned.

Table 7: Architecture optimizer settings on PDE tasks. Note that the step-size is updated using the same schedule as the backbone.

| task | optimizer | initial step-size | warmup epochs |
|---|---|---|---|
| 1d Burgers' equation | Adam | 1E-3 | 0 |
| 1d Burgers' equation (FNO init) | Momentum(0.5) | 1E-4 | 250 |
| 2d Darcy Flow | Momentum(0.5) | 1E-1 | 0 |
| 2d Darcy Flow (FNO init) | Momentum(0.5) | 1E-1 | 0 |
| 2d Navier Stokes ($\nu = 10^{-4}, T = 30$) | Momentum(0.5) | 5E-3 | 0 |
| 2d Navier Stokes ($\nu = 10^{-5}, T = 20$) | Momentum(0.5) | 1E-3 | 0 |

Table 8: Test relative errors on the 1d Burgers' equation. We were not able to match the FNO-1d results reported by the authors [30] using their published codebase, however, our proposed XD operations outperform our reproduction of their results at every resolution. Furthermore, we outperform their reported test relative errors on every resolution except $s = 4096$, where we roughly match their performance.

| Method (source) | $s = 256$ | $s = 512$ | $s = 1024$ | $s = 2048$ | $s = 4096$ | $s = 8192$ |
|---|---|---|---|---|---|---|
| NN [30] | 0.4714 | 0.4561 | 0.4803 | 0.4645 | 0.4779 | 0.4452 |
| GCN [30] | 0.3999 | 0.4138 | 0.4176 | 0.4157 | 0.4191 | 0.4198 |
| FCN [30] | 0.0958 | 0.1407 | 0.1877 | 0.2313 | 0.2855 | 0.3238 |
| PCANN [30] | 0.0398 | 0.0395 | 0.0391 | 0.0383 | 0.0392 | 0.0393 |
| GNO [30] | 0.0555 | 0.0594 | 0.0651 | 0.0663 | 0.0666 | 0.0699 |
| LNO [30] | 0.0212 | 0.0221 | 0.0217 | 0.0219 | 0.0200 | 0.0189 |
| MGNO [30] | 0.0243 | 0.0355 | 0.0374 | 0.0360 | 0.0364 | 0.0364 |
| FNO-1d [30] | 0.0149 | 0.0158 | 0.0160 | 0.0146 | **0.0142** | 0.0139 |
| CNN (ours) | 0.0518 | 0.1220 | 0.1830 | 0.2280 | 0.2730 | 0.2970 |
| FNO-1d (reproduced) | 0.0181 | 0.0191 | 0.0188 | 0.0184 | 0.0183 | 0.0183 |
| CNN XD (ours) | **0.0141** | **0.0079** | **0.0154** | **0.0099** | 0.0145 | **0.0123** |
| FNO-1d XD (ours) | 0.0153 | 0.0154 | 0.0154 | 0.0167 | 0.0160 | 0.0155 |

Table 9: Test relative errors on 2d Darcy Flow. Our reproduction of the FNO-2d results outperform those reported by the authors [30]. Nonetheless, our proposed XD operations outperform both our reproduction and the reported results at every resolution.

| Method (source) | $s = 85$ | $s = 106$ | $s = 141$ | $s = 211$ | $s = 421$ |
|---|---|---|---|---|---|
| NN [30] | 0.1716 | - | 0.1716 | 0.1716 | 0.1716 |
| GCN [30] | 0.0253 | - | 0.0493 | 0.0727 | 0.1097 |
| FCN [30] | 0.0299 | - | 0.0298 | 0.0298 | 0.0299 |
| PCANN [30] | 0.0244 | - | 0.0251 | 0.0255 | 0.0259 |
| GNO [30] | 0.0346 | - | 0.0332 | 0.0342 | 0.0369 |
| LNO [30] | 0.0520 | - | 0.0461 | 0.0445 | - |
| MGNO [30] | 0.0416 | - | 0.0428 | 0.0428 | 0.0420 |
| FNO-2d [30] | 0.0108 | - | 0.0109 | 0.0109 | 0.0098 |
| CNN (ours) | 0.0404 | 0.0495 | 0.0613 | 0.0813 | 0.1150 |
| FNO-2d (reproduced) | 0.0096 | 0.0092 | 0.0091 | 0.0091 | 0.0091 |
| CNN XD (ours) | **0.0065** | **0.0065** | **0.0065** | **0.0071** | **0.0066** |
| FNO-2d XD (ours) | 0.0082 | 0.0079 | 0.0077 | 0.0076 | 0.0074 |

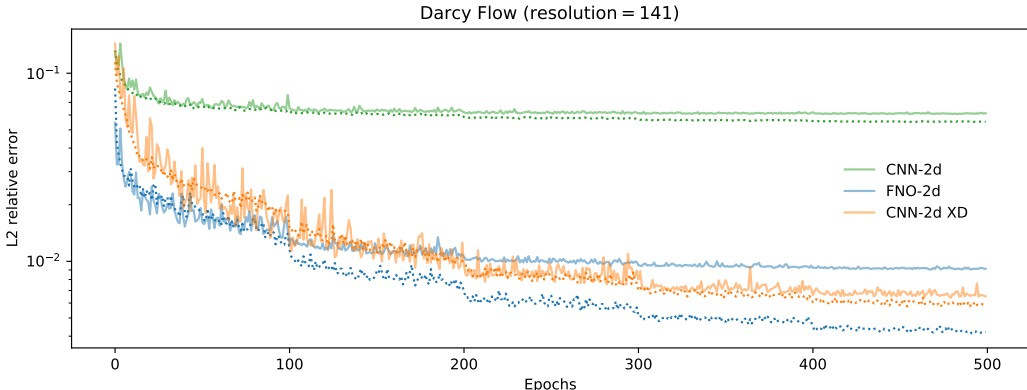

Figure 5: Training curves (dotted) and test curves (solid) on Darcy Flow at resolution 141, showing better generalization of XD-operations.

# E  Experimental Details: Protein Folding

Table 10: Architecture optimizer settings on for our protein folding experiments, across different ResNet depths. Note that the same step-size is used throughout since the backbone has no step-size schedule.

| search space | optimizer | step-size | warmup epochs |
|---|---|---|---|
| ResNet-4 XD | Adam | 1E-4 | 2 |
| ResNet-6 XD | Momentum(0.99) | 1E-4 | 2 |
| ResNet-10 XD | Momentum(0.99) | 1E-3 | 2 |
| ResNet-18 XD | Momentum(0.9) | 5E-4 | 2 |
| ResNet-34 XD | Momentum(0.9) | 5E-4 | 2 |

Table 11: Test MAE$_8$ of the Dilated ResNet of [1], compared to a standard ResNet backbone and XD-operations applied to ResNet. Results are averaged over 3 trials.

| Method | depth $= 4$ | depth $= 6$ | depth $= 10$ | depth $= 18$ | depth $= 34$ |
|---|---|---|---|---|---|
| ResNet | $5.99 \pm 0.43$ | $5.30 \pm 0.11$ | $4.91 \pm 0.25$ | $4.80 \pm 0.07$ | $4.66 \pm 0.15$ |
| Dilated ResNet | $6.04 \pm 0.33$ | $5.49 \pm 0.02$ | $4.64 \pm 0.08$ | $4.59 \pm 0.22$ | $4.50 \pm 0.13$ |
| ResNet XD | $\mathbf{5.59 \pm 0.09}$ | $\mathbf{4.59 \pm 0.17}$ | $\mathbf{4.25 \pm 0.16}$ | $\mathbf{4.22 \pm 0.03}$ | $\mathbf{4.00 \pm 0.07}$ |

For our protein folding experiments, our code is a PyTorch re-implementation of the PDNET code and setup [1] provided here: `https://github.com/ba-lab/pdnet`. As before, we use the same training routine and settings as the Dilated ResNet architecture and only tune the architecture optimizer. We consider the following hyperparameters for the architecture optimizer: Adam vs. SGD (with or without momentum), learning rate, and number of warmup epochs. The final hyperparameters for each depth can be found in Table 10. Our ResNet backbone differs from Dilated ResNet in that its dilation rate is set to 1 in every convolutional layer. In Table 11, we present average $MAE_8$ on the PSICOV test set for each method at each depth.

## F   Experimental Details: Music Modeling and Sequence Modeling

Table 12: Architecture optimizer settings on sequence modeling tasks. Note that the step-size is updated using the same schedule as the backbone.

| task | optimizer | initial step-size | warmup epochs |
|---|---|---|---|
| Permuted MNIST | Adam | 2E-4 | 0 |
| JSB Chorales | Adam | 2E-4 | 25 |
| Nottingham | Adam | 2E-3 | 0 |
| Penn Treebank | Adam | 2E-6 | 0 |

Table 13: XD-operations applied to TCNs compared to recent empirical results in sequence modeling. Our results are averages of three trials. Methods achieving within one deviation of the best performance are **bolded**.

| Method (source) | Permuted MNIST* (error) | JSB Chorales (loss) | Nottingham (loss) | Penn Treebank (perplexity) |
|---|---|---|---|---|
| LSTM [5] | 14.3 | 8.45 | 3.29 | 78.93 |
| GRU [5] | 12.7 | 8.43 | 3.46 | 92.48 |
| RNN [5] | 74.7 | 8.91 | 4.05 | 114.50 |
| TCN backbone [5] | 2.8 | 8.10 | 3.07 | 88.68 |
| TrellisNet [6] | 1.87 | - | - | **54.19** |
| R-Transformer [44] | - | - | **2.37** | 84.38 |
| HiPPO-LegS [14] | **1.7** | - | - | - |
| TCN backbone (reproduced) | $2.89 \pm 0.04$ | $8.17 \pm 0.01$ | $2.97 \pm 0.01$ | $88.49 \pm 0.31$ |
| TCN backbone XD (ours) | $\mathbf{1.75 \pm 0.11}$ | $\mathbf{8.07 \pm 0.02}$ | $2.81 \pm 0.05$ | $84.11 \pm 0.25$ |
| Undilated TCN (ours) | $11.3 \pm 2.1$ | $8.16 \pm 0.04$ | $3.21 \pm 0.02$ | $94.30 \pm 0.33$ |
| Undilated TCN XD (ours) | $\mathbf{1.77 \pm 0.10}$ | $\mathbf{8.07 \pm 0.01}$ | $2.84 \pm 0.02$ | $85.04 \pm 0.49$ |

*We use depth $\mathbf{d} = (3, 3, 3)$ XD-operations for permuted MNIST experiments; elsewhere we use $(1, 3, 1)$. Results within a standard deviation of the best are **bolded**.

For our sequence modeling experiments we use the TCN code [5] provided here: `https://github.com/locuslab/TCN`. As before we use the same settings and training routine as the backbone for all tasks, tuning only the architecture optimizer. The specific settings are provided in Table 12. For both the baselines and XD-operations we use the same optimizer settings for both the dilated and undilated TCN backbones. In Table 13 we present results for both music modeling and for two additional benchmarks—permuted MNIST and Penn Treebank—on which we see a similar pattern of XD-operations being able to recover and even beat (dilated) TCN performance starting from an undilated network.