# OpenReview forum: "Rethinking Neural Operations for Diverse Tasks"
_NeurIPS.cc/2021/Conference — NeurIPS 2021 Poster_

### Official Review · Reviewer_nncs · 2021-07-01

**Rating:** 7
**Confidence:** 4

**Summary:**

The paper starts by introducing Kaleidoscope-matrices (K-matrices). The core idea here is:
1. Most of the interesting operations in NAS spaces are linear and can be expressed as a matrix multiplication $A_W x$
2. The operations can therefore be diagonalized by a DFT
3. We can factorize the DFT matrices into “butterfly matrices” (such that, for a factorization of factor $N$, we can write $F =$ the product of the $N$ butterfly matrices and a permutation matrix).
4. By allowing this factor $N$ to vary, the matrix multiplication $A_W x$ can be generalized to express a wider family of operations

The core contribution of this paper appears to be using K-matrices in a NAS setting (which involves a reasonable amount of work):
- The architecture parameters are continuous
- K-matrices can’t express parameter-free ops
- Finding/evaluating new operations in a NAS setting

Their NAS process then looks like:
1. Take in a backbone and generalize it to a supernet
2. Simultaneously train the architecture parameters and the weights
3. Extract a net + weights for inference

The paper then explores a few unusual tasks and shows that this generalization process is worthwhile.


**Ethical Concerns:**

I have no ethical concerns with this paper.

**Limitations And Societal Impact:**

The authors addressed the potential negative societal impact of their work.

In general, moving NAS away from vision and language tasks is a direction that we should probably encourage.

**Main Review:**

Overall this paper was a joy to read, filled with insightful interpretations of things I previously thought boring (e.g. the definition of NAS spaces as a set of operations with the same W,X,Y and their expressibility). The paper would be strengthened by more thorough empirical results, but I understand why in this particular setting that could hard to provide.

The paper has a hard task: construct a general framework for neural architecture search that will let us do more exciting things than squeeze out performance on existing benchmarks, but that also doesn’t perform drastically worse than existing (extremely heavily engineered) techniques on said benchmarks.

I think the authors do a good job of showing the expressivity of their method with regards to diverse tasks while also showing that it continues to work on the more boring tasks (e.g. CIFAR-10). Even so, I do think the lack of focus on the typical “NAS-bench” setups is a little disappointing. I would have liked to see what happened if the authors took a NAS-bench cell/skeleton as their starting point — since we have results for a wide range of NAS-techniques on those benchmarks, it would be easier to put their method in context.

The authors perform some analysis of the Euclidean distance between their architecture parameters (which I’m assuming is butterfly factors + bias?) and the traditional convolution operation show that they can tailor then operations to suit the complexity of the tasks, which is very nice.

The authors then dive into a few diverse tasks, none of which I have experience with so may have missed some evaluation points:

**Solving PDEs**: this section is neat, and it shows that the XD-operations can be used to replace the hand-engineering effort that went into designing Fourier Neural Operators. I may have missed it but it would be good to have some detail on the dataset specifically (number of training/test samples per PDE/resolution/viscosity). My immediate concern is that learning over a space of operations like this probably requires a lot of data?

**Protein folding distance prediction**: I don’t know much about this task but the results seem good (there are minor improvements over both ResNet and Dilated ResNet at each depth).

**Music modeling**: again, I don’t know much about the task but the results seem good.

I’m happy that the evidence provided is sufficient to show that this process can be used to take a backbone, generalize it to some family of ops, and then specialize the operations to suit the task. That alone is a very useful contribution.

In the abstract the authors state “Starting with any standard backbone such as ResNet, we show how to transform it into a search space over XD-operations and how to traverse the space using a simple weight-sharing scheme.” My assumption from this sentence was that they would use is a single backbone used for all three of the tasks. That they don’t suggests that though the method provides a good level of generalization, it still requires some hand engineering to get it to work well.

## Overview

**Originality**: In some ways this felt like the inevitable next step to [1]. However, it appears there is still a reasonable amount of novelty/originality to the paper in getting K-matrices to work for NAS. Related work is adequately cited.

**Quality**: The submission is technically sound, and filled with satisfying definitions of the various aspects of the problem. The authors show the performance of their method on three unusual problems, plus the more classic task of CIFAR-10. The step from CIFAR $\rightarrow$ permuted CIFAR $\rightarrow$ other tasks is a nice logical narrative. The experimental evidence isn't overwhelmingly strong, but the paper does break NAS into relatively new territory so I don't think a huge amount of empirical evaluation is needed.

**Clarity**: The submission is very clearly written, organized, and thoroughly cites background material which was useful for me as a reader with not a great deal of knowledge regarding butterfly matrices and the related work. It's an interesting topic because the paper is trying to generate new types of operation which by their nature probably won't be particularly intuitive, but I feel like the authors did a good job of showing why a few example operations are expressible in their search space.

**Significance**: The paper addresses a major limitation with neural architecture search: that we typically only use a very restrictive set of operations that we know to work well on a limited range of tasks. I'm not convinced that after this paper, everybody in the NAS subfield will switch to using K-matrices, but it seems like a useful tool in the toolbox for people who are using NAS outside of CIFAR-10.

## Questions

1. I’m a little confused about exactly what the “architecture parameters” look like. Is it a collection of “butterfly factors” and biases for K, L, and M? Are all of the cells in the network the same or do you have different architecture parameters for each layer?
2. In my interpretation, the authors are almost implicitly proposing a new “NAS-bench” which is composed of XD-operations (a much wider family of operations than what we currently use). To that end, it would be nice to see some statistics about what the space of XD-operations looks like: if you randomly sample from the space, how much variance is there in the performance of the operations? Is the space dense with very-convolution-like operations or is it quite uniformly distributed? What happens if you take the cell-structure/backbone from NATS-bench/ NAS-Bench-101/ NAS-Bench-201 and set all of the operations to be XD-operations?
3. On page 6 the authors state “On CIFAR-10 we do not expect to see much improvement from XD-operations over the CNN backbone used to initialize search, as convolutions are already the “right” operation for images.” I guess this is true, but I was wondering whether the authors think there is any scope to improve on convolution using their method in future? Given the volume of work on convolutional-alternatives (LambdaNetworks, the Shift operation, OctaveConvolution etc.) for image domains, I was hoping that within a space of XD-Operations we might find more of these.
4. One thing I can’t quite figure out is whether all of the operations expressible in the space of XD-Operations are reduced forms of convolution (e.g. adding dilation)?  Could you express something like transposed convolution? Is there anything expressible as an XD-Operation that isn’t expressible as unstructured pruning in the matrix A?
5. An evaluation of the limitations of the expressivity of the search space would also be nice. For example, I assume you probably can’t express something like a self-attention layer due to the softmax?

[1] Dao, Tri, et al. "Kaleidoscope: An efficient, learnable representation for all structured linear maps." arXiv preprint arXiv:2012.14966 (2020).


**Time Spent Reviewing:**

4

---

> ### Author Response · Authors · 2021-08-11
> **Response to Reviewer nncs**
>
> Thank you for your positive review. We are glad you appreciated some of the detailed discussions, e.g. on expressivity, and hope to address some of your other questions and concerns below.
>
> ### Responses to enumerated questions:
> 1\. [*I’m a little confused about exactly what the “architecture parameters” look like. Is it a collection of “butterfly factors” and biases for K, L, and M? Are all of the cells in the network the same or do you have different architecture parameters for each layer?*]
> - Yes, the main architecture parameters are a collection of butterfly factors for K, L, and M (these have no biases). There is also a bias b and channel gates C. We have different architecture parameters for each layer; in-fact we found that forcing them to be the same led to poor performance due to training difficulty. We will make sure to note this in revision.
>
> 2\. [*In my interpretation, the authors are almost implicitly proposing a new “NAS-bench” which is composed of XD-operations (a much wider family of operations than what we currently use). To that end, it would be nice to see some statistics about what the space of XD-operations looks like: if you randomly sample from the space, how much variance is there in the performance of the operations? Is the space dense with very-convolution-like operations or is it quite uniformly distributed? What happens if you take the cell-structure/backbone from NATS-bench/ NAS-Bench-101/ NAS-Bench-201 and set all of the operations to be XD-operations?*]
> - It is not necessarily clear which distribution to use when sampling from the search space. One approach is to set the K, L, and M matrices to be random unitary matrices, which we found leads to very poor performance on both CIFAR and permuted CIFAR. Indeed it seems that the “inductive bias” for search provided by initializing as vanilla convolutions---whereby K, L, and M are Fourier matrices---is useful even when vanilla convolutions are clearly suboptimal operations. This combined with the fact that XD-operations performed well with multiple different CNN backbones throughout our paper suggests that XD-operations will perform reasonably well when combined with any (non-degenerate) backbone from the NAS-benches. We view combining XD with search over such traditional spaces as a likely fruitful direction for making the entire process even more automated.
>
> 3\. [*On page 6 the authors state “On CIFAR-10 we do not expect to see much improvement from XD-operations over the CNN backbone used to initialize search, as convolutions are already the “right” operation for images.” I guess this is true, but I was wondering whether the authors think there is any scope to improve on convolution using their method in future? Given the volume of work on convolutional-alternatives (LambdaNetworks, the Shift operation, OctaveConvolution etc.) for image domains, I was hoping that within a space of XD-Operations we might find more of these.*]
> - XD-operations may indeed contain some convolutional alternatives; indeed, from a look at the variants you mention it is likely Shift operations and OctaveConvolutions are contained in straightforward extensions of our search space (LambdaNetworks seem nonlinear and thus would require more work). Perhaps a better search procedure than the fairly simple one we use could find such operations and also lead to better performance on CIFAR. Note also that a major goal of many of these operations, e.g. Shift and OctConv, seems to be to improve the computational efficiency of convolutions, which our method does not aim to do; thus a better search method over XD-operations would also likely require some approaches involving constrained or multi-objective NAS. We believe this is a valuable direction for future work.
>
> 4\. [*One thing I can’t quite figure out is whether all of the operations expressible in the space of XD-Operations are reduced forms of convolution (e.g. adding dilation)? Could you express something like transposed convolution? Is there anything expressible as an XD-Operation that isn’t expressible as unstructured pruning in the matrix A?*]
> - It is true that many of the XD-operations that we highlight are variants of convolutions, including dilated convolutions and average pooling. In fact even FNO can be viewed as a convolution with a complex kernel. However, the vastness of the set of K-matrices means most operations are not convolutions; just as a start, any permutation followed by a convolution is an XD-operation but is not necessarily a convolution, and there are combinatorially many possible permutations. In fact, transposed convolution can be expressed as a permutation followed by a convolution, so yes, our search space contains that operation (thank you for this suggestion). As for your last question - yes, the matrix A resulting from XD-operations is not necessarily sparse; as the simplest example, simply applying the Fourier transform is an XD-operation, but the DFT matrix is not sparse.
>
> 5\. [*An evaluation of the limitations of the expressivity of the search space would also be nice. For example, I assume you probably can’t express something like a self-attention layer due to the softmax?*]
> - We agree and will include more such examples in revision (we currently mention only Max-Pooling). You are correct that XD-operations do not contain self-attention, not only because of the softmax but also because of the quadratic interaction terms. Another limitation is BatchNorm.
>
> ### Responses to some other comments
>
> 6\. [*The paper would be strengthened by more thorough empirical results, but I understand why in this particular setting that could hard to provide.*]
> - You may be interested in some additional results we included in our response to Reviewer misB (point 2) that we will add in revision.
>
> 7\. [*I may have missed it but it would be good to have some detail on the dataset specifically (number of training/test samples per PDE/resolution/viscosity). My immediate concern is that learning over a space of operations like this probably requires a lot of data?*]
> - We use the same datasets and training procedures proposed in the original papers from which we obtained our tasks. Following the FNO paper [28], we use 1000 training examples and 100 testing examples for each resolution of the 1d Burgers’ equation and the 2d Darcy Flow problems. For the Navier Stokes problems, we use 1000 training examples and 200 testing examples for each viscosity. The PDNET dataset for protein folding is similarly small, with 3356 proteins used for training, 100 validation proteins, and 150 proteins in the test set. This is deliberately smaller than other protein folding benchmarks so as to fill a need for a “small and representative dataset packaged for faster development and testing.” Our results show that XD does well on all of these tasks despite training on relatively small datasets.
>
> 8\. [*In the abstract the authors state “Starting with any standard backbone such as ResNet, we show how to transform it into a search space over XD-operations and how to traverse the space using a simple weight-sharing scheme.” My assumption from this sentence was that they would use is a single backbone used for all three of the tasks. That they don’t suggests that though the method provides a good level of generalization, it still requires some hand engineering to get it to work well.*]
> - We had a choice here between using the same network for all three applications or using networks used in past work on the applications considered, and we chose the latter in order to facilitate more direct comparison. On top of this, the only engineering required was tuning settings of the architecture optimizer. However, it is likely the case that past authors chose these networks over others for a reason, so in some sense they did the hand engineering for us. To make this process more automated in-practice, one would likely combine XD with a more traditional NAS algorithm, similar to your suggestion concerning NAS-benches.

---

> > ### Comment · Reviewer_nncs · 2021-08-17
> > **Response to authors**
> >
> > Thanks to the authors for their detailed response.
> >
> > It seems that most of my questions were really just converging towards combining this technique with some form of traditional NAS algorithm, and I think the authors have clearly demonstrated that XD-operations are expressive enough to offer a broad scope of potential future work.
> >
> > **I won't raise my score because:** in order to be a really exceptional paper, I think I would have had to have seen some stronger empirical evidence in that direction. The results are interesting, but I don't believe there's anything here that will have an immediate drastic effect on modern neural architectures.
> >
> > **I won't lower my score because:** I think the authors have done a good enough job illustrating that generating diverse operations can help us tackle more exciting tasks and move beyond standard architectures and operations.
> >
> > In other words I'm very excited about future work in this direction, but see this paper as a crucial first step rather than a panacea.
> >
> > I have one last question for the authors:
> >
> > > One approach is to set the K, L, and M matrices to be random unitary matrices, which we found leads to very poor performance on both CIFAR and permuted CIFAR
> >
> > OK, this sounds reasonable. One thing I'm curious about is whether the randomly sampled factors give operations with large variance, or whether they are just uniformly poor?

---

> > > ### Author Response · Authors · 2021-08-19
> > > **Follow-up response to Reviewer nnc**
> > >
> > > Thank you for your feedback. A short answer to your last question is "uniformly poor." In more detail, we ran 10 seeds where we fixed K, L, and M to be random unitary matrices on three of the tasks in the paper. The mean performance and standard deviation of these seeds are in the "XD (random)" row of the following table, which compares them to results already in the paper:
> > >
> > > |               | CIFAR-10    | Permuted | Darcy Flow (resolution=85)     |
> > > |---------------|----------|----------------|----------------|
> > > | CNN           | 91.7±0.2 | 58.6±0.7       | 0.0405±0.0003  |
> > > | XD (searched) | 92.4±0.2 | 73.5±1.6       | 0.00692±0.0003 |
> > > | XD (random)   | 63.0±0.3 | 55.6±0.3       | 0.0912±0.002   |
> > > | FNO           | N/A      | N/A            | 0.0105±0.0002  |
> > >
> > > The variation in performance of the randomly sampled architectures is not significantly higher than the variation due to randomness in the training process, except on Darcy Flow where it is an order of magnitude higher (but still not large). The random architectures perform the worst on all three tasks, although they are fairly close to CNNs on Permuted CIFAR. This means it is unlikely that we can obtain a good architecture by randomly sampling from this distribution, meaning that any search algorithm must either be more sophisticated than random search or be using some better distribution (note that CNNs are in the support of this distribution because DFTs are unitary; XD (searched) architectures are not, because we do not constrain search to unitary K-matrices).

---

### Official Review · Reviewer_9aTC · 2021-07-11

**Rating:** 3
**Confidence:** 4

**Summary:**

The core contribution of this work is to re-image of NAS operation spaces to include both standard operations as well as new operations. This work replaces the Discrete Fourier Transform (DFT) in the diagonal decomposition by a more expressive family of efficient liner transforms known as Kaleidoscope or K-matrices (Expressive Diagonalization Operations), which comprises of a large search space of various types of grid-based convolutions and pooling, permutations, graph convolutions, the Fourier Neural Operator, and more. This work developed a simple procedure which transforms any backbone convolutional neural network into an architecture search space by replacing its operations with XD-operations. This space is then searched using a simple weight-sharing algorithm. This work yields models that are 15% more accurate than standard discrete search spaces on permuted CIFAR-10, highlighting the fragility of standard NAS operation spaces on new datasets, and thus the need for XD-operations.

**Limitations And Societal Impact:**

Yes, both are addressed.

**Main Review:**

This work brings up an interesting perspective that ,. This work also proposes to replace matrices F and F-1 in by any one of a general family of efficient matrices, which yields the single-channel version of expressive diagonalization operations.

Strengths:
1. This approach in general provides a much more expressive search space for NAS, by devising XD-opeartions with K-matrix architecture parameters.
2. The formulation and complexity analysis on a single-channel XD-operations with K-matrix architecture parameters is pretty clear.

Weakness:
1. The paper is not clear about how to rank models or evaluate models generated from this enlarged search space. The reviewer is particularly confused by "Note that the fully continuous nature of XD-operations means that we will only evaluate the final network returned by search".

2. Empirical results are weak. Only Cifar10 results are provided and only bad baselines (LeNet and ResNet-20) are compared against. Also, it is odd that "On CIFAR-10 we do not expect to see much improvement from XD-operations over the CNN backbone used to initialize search, as convolutions are already the “right” operation for images", why this work should evaluate on CIFAR10 then?

5. Application section is interesting but not very formal.  There isn't any standard benchmarks used or any well-known baselines are compared upon.

Overall, the reviewer thinks the general direction is interesting, however, evaluation is poor. The work is not thoroughly evaluated and no strong baselines are provided.


**Time Spent Reviewing:**

2

---

> ### Author Response · Authors · 2021-08-11
> **Response to Reviewer 9aTC**
>
> Thank you for your review. We respectfully disagree with the three weaknesses you discuss; in particular, the focus on state-of-the-art comparisons for CIFAR and the request for standard benchmarks/well-known baselines in the three application areas seem to stem from a possible misunderstanding of the goal of the work. We hope to clarify these points in our response below. In particular, we emphasize that, as stated on lines 35-36, our claimed contribution is that we “take critical steps towards a broader NAS that enables the discovery of good design patterns with limited human specification from data in under-explored domains.” Thus the evaluation of our contribution should take into account that under-explored applications may not have standard benchmarks, and that CIFAR, a task in a well-explored domain, should be treated as an illustrative example and not a target task.
>
> ### Responses to concerns
> 1.1 [*The paper is not clear about how to rank models or evaluate models generated from this enlarged search space.*]
> - Architectures in the search space can be evaluated in the standard way: how well they perform on the appropriate validation metric (e.g. accuracy, loss) when combined with a set of model weights. However, since the search space is continuous, with architecture parameters specified by real values, it is difficult to report rank correlation metrics evaluating how well the discovered shared model weights perform over the entire search space, as this usually involves sampling from the uniform distribution over architectures; in our case there is no obvious distribution over XD-operations.
>
> 1.2 [*The reviewer is particularly confused by "Note that the fully continuous nature of XD-operations means that we will only evaluate the final network returned by search".*]
> - Unlike common NAS algorithms such as DARTS, which must discretize the supernet returned by their continuous search procedure, in our case the supernet architecture will already be in the search space and so we will evaluate it directly.
>
> 2\. [*Empirical results are weak. Only Cifar10 results are provided and only bad baselines (LeNet and ResNet-20) are compared against. Also, it is odd that "On CIFAR-10 we do not expect to see much improvement from XD-operations over the CNN backbone used to initialize search, as convolutions are already the “right” operation for images", why this work should evaluate on CIFAR10 then?*]
> - We respectfully disagree with this assessment. As stated on line 248, the CIFAR example is a warmup and not the main objective of our work; instead, the three application sections contain the main empirical results that should be evaluated. The CIFAR example is used to do two things: (1) illustrate how to use our search space using a dataset with which most readers will be familiar and (2) demonstrate how fragile discrete NAS approaches are to non-image datasets by evaluating architectures on permuted CIFAR. Since this is a toy example, we use toy networks in our analysis in the main paper. However, in Table 6 of the Appendix we do compare directly to much stronger networks, including architectures discovered by the state-of-the-art NAS algorithms DenseNAS [13] and GAEA PC-DARTS [23]. These results tell a similar story: standard NAS algorithms discover good networks for CIFAR but perform much worse than XD on permuted CIFAR.
>
> 3\. [*Application section is interesting but not very formal. There isn't any standard benchmarks used or any well-known baselines are compared upon.*]
> - As stated in the introduction, the main goal of this work is to enable NAS in under-explored domains beyond vision and text. This means that standard benchmarks will not be available by definition, since nearly all NAS benchmarks focus on popular domains. Furthermore, in all three applications we do compare to strong baselines: architectures that use operations that were hand-designed by human experts to solve these tasks. For example, for PDE solvers we compare to FNO, which is state-of-the-art as of ICLR 2021 [28], while for music modeling we compare to the TCN network from a well-known benchmarking paper with more than a thousand citations on Google Scholar [5]. Since the goal of NAS is to automate-away the need for human experts, such baselines are the right comparators in our setting, and we believe our approach compares favorably. Finally, note that in our response to Reviewer misB (Question 2) we have included further comparisons to more traditional search spaces in several settings where it is most feasible.

---

> > ### Comment · Reviewer_9aTC · 2021-08-17
> > **Response to authors**
> >
> > Thank you for elaborating on the questions.
> >
> > Being a practitioner, the reviewer believes that being 10x better than baselines on unimportant or non-challenging tasks is less good compared to being 1.1x better on important and challenging tasks (e.g. ImageNet, COCO, GLUE, etc.). Therefore, the reviewer holds the concerns that a cute idea cannot be executed in real if the method is not thoroughly evaluated on more important and relevant tasks.
> >
> > I won't raise my score because: 1. Agree with Reviewer nncs, I don't believe there's anything here that will have an immediate drastic effect on modern neural architectures. 2. The same formulation does not need to applied to NAS, but can be applied to a stronger state-of-the-art neural architecture (that has demonstrated SoTA performance on ImageNet or some more important, well-known tasks) and outperforms it.

---

> > > ### Author Response · Authors · 2021-08-18
> > > **Follow-up response to Reviewer 9aTC**
> > >
> > > Thank you for clarifying your position. We believe your evaluation reflects views that are not shared by the broader ML community. In particular:
> > >
> > > 1\. [*the reviewer believes that being 10x better than baselines on unimportant or non-challenging tasks is less good compared to being 1.1x better on important and challenging tasks (e.g. ImageNet, COCO, GLUE, etc.)*]
> > > - We strongly disagree that achieving short-term incremental gains on well-known/large-scale problems is the only way to do research, and believe that it is also possible to make valuable contributions by aiming for long-term impact and opening new directions. This statement also dismisses as “unimportant or non-challenging” the non-vision applications we consider, which include (1) a PDE-solving task that a [well-received](https://www.technologyreview.com/2020/10/30/1011435/ai-fourier-neural-network-cracks-navier-stokes-and-partial-differential-equations/) ICLR 2021 [paper](https://openreview.net/forum?id=c8P9NQVtmnO) showed to be difficult for standard CNNs and (2) a variant of a protein folding task that has been a [long-standing challenge in computer science](https://www.theguardian.com/technology/2020/nov/30/deepmind-ai-cracks-50-year-old-problem-of-biology-research) and which [gained significant attention](https://deepmind.com/blog/article/alphafold-a-solution-to-a-50-year-old-grand-challenge-in-biology) in the ML community last year.
> > >
> > > 2\. [*the reviewer holds the concerns that a cute idea cannot be executed in real if the method is not thoroughly evaluated on more important and relevant tasks*]
> > > - The main goal of this work is to enable NAS in under-explored domains beyond vision and text, so the important and relevant tasks in its evaluation must be from challenging problems beyond vision and text tasks. While the reviewer has described these tasks as “unimportant or non-challenging,” we believe this subjective view is not shared by the broader ML community.
> > >
> > >
> > > 3\. [*Agree with Reviewer nncs, I don't believe there's anything here that will have an immediate drastic effect on modern neural architectures.*]
> > > - While we do not disagree with the statement, from the history of past papers at NeurIPS it is clear that most researchers do not view it as necessary for a paper to have “immediate drastic impact” in practice to be impactful (and as noted above, we feel that in general this is a very restrictive viewpoint on research impact). Moreover, citing agreement with a single point by another reviewer while omitting the massive difference in the overall evaluation paints a very misleading picture (Reviewer nncs summarizes that they are “very excited about future work in this direction, but see this paper as a crucial first step rather than a panacea”).
> > >
> > > 4\. [*The same formulation does not need to applied to NAS, but can be applied to a stronger state-of-the-art neural architecture (that has demonstrated SoTA performance on ImageNet or some more important, well-known tasks) and outperforms it.*]
> > > - Similar to the second point, we seem to fundamentally disagree with the necessity of this. Additional experiments (beyond those already in the paper) focused on SOTA networks for vision/text tasks are orthogonal to the goal of this work, which is to enable NAS on tasks beyond vision/text.

---

### Official Review · Reviewer_misB · 2021-07-14

**Rating:** 7
**Confidence:** 2

**Summary:**

This paper explores learning more general neural architectures by leveraging a novel, continuous parameterization of architecture (based on kaleidoscope marix) and one-shot style NAS algorithms. They present results on standard, and permuted cifar10, then 3 more exotic datasets including PDE solutions, problems from protein folding, and music modeling.

**Limitations And Societal Impact:**

Some limitations are discussed, but as noted above I would appreciate some discussion on inference costs / how to actually use these models / some notion of overhead at inference time.

No societal impact is discussed.

**Main Review:**

This seems like a solid paper. I would advocate for accepting though I am not an expert in this field (hence my low confidence). My rational for a 6 vs 7 is the limited baselines comparing to existing NAS methods.

Clarity: My main concern is about clarity of the presentation. I found the the abstract discussion based solely on equations of this method difficult to follow. I realize the paper is space constrained, but including examples of K-matrix / even showing how standard operations map into XD-operators would be greatly appreciated. The description of the method also takes around 5-6 pages as well which feels very long.

Results:
I found these experiments quite compelling. Having baselines that outperform existing work is always great!

For the PDE, protein and music, it would be interesting to see another NAS comparison -- in particular a discrete search space style NAS algorithm like what was done in the cifar10 experiments.

One question I have is about how useful the resulting architectures are. In Darts, one can discretize the discrete search space resulting in very efficient architectures for future tasks. In this work, this doesn't seem so straight forward (and it is even stated that this was not done on purpose). Do the K-matrix style formulation lead to an efficient formula? Naively it seems like without it it would be up to 10x more costly (judging by the resnet-20 cifar10 walltimes).

Related, but do you have any understanding of what kinds of "architectures" these things learn? Is anything interpretable?






**Time Spent Reviewing:**

2

---

> ### Author Response · Authors · 2021-08-11
> **Response to Reviewer misB**
>
> Thank you for your positive review. We hope to address your questions below. In particular, we aim to address your main concern---a lack of NAS baselines for the application sections similar to those provided for the CIFAR example---by providing them for the applications that use 2d data (which most popular NAS search spaces require). We will include the results in revision, specifically by replacing Figure 3 (right) and Figure 4 by new plots for the Darcy Flow PDE ([link](https://drive.google.com/file/d/1t-rK4psYah2uWcuHeypx_NRc172OsZbg/view?usp=sharing)) and PSICOV protein ([link](https://drive.google.com/file/d/1NPTwxhW021AMQXSoNnx3jiu55rHm-x9Y/view?usp=sharing)) tasks, respectively. In both cases the baseline of using DARTS operations in the vanilla CNN backbone usually underperforms XD. We also compare
>  to Auto-DeepLab ([Liu et al., CVPR 2019](https://arxiv.org/abs/1901.02985)), a well-known NAS method specifically designed for such dense prediction tasks, and found that it is significantly outperformed by XD on both 2d tasks. Both evaluations reinforce our existing results with more direct NAS comparisons.
>
> ### Responses to questions/suggestions
> 1\. [*I found the the abstract discussion based solely on equations of this method difficult to follow. Including examples of K-matrix / even showing how standard operations map into XD-operators would be greatly appreciated.*]
> - With the extra page allowed for a revision we hope to include an example of a K-matrix, although note that this a contribution of past work [10]. As for showing how standard operations map into XD-operations: we agree that this is very useful and do show this for a few operations in the main paper, including skip-connections (line 173), average pooling (line 174), and dilated convolutions (Equation 6). We go into further detail for many more operations---including other types of convolutions, graph convolutions, and FNOs---in Appendix A. Please let us know if these examples are useful or if you believe some of the appendix examples should be in the main text.
>
> 2\. [*For the PDE, protein and music, it would be interesting to see another NAS comparison -- in particular a discrete search space style NAS algorithm like what was done in the cifar10 experiments.*]
> - Thank you for this suggestion; we agree that such a comparison is useful. In revision we will include additional evaluations of the DARTS operations that we ran on the Darcy Flow PDE task and the PSICOV protein task; these were chosen because they have 2d data and are thus suitable for existing search spaces. For DARTS operations, the same procedure as for the CIFAR-10 experiments was used, with the architecture search procedure of DARTS being tuned separately for each task. For Auto-DeepLab ([Liu et al., CVPR 2019](https://arxiv.org/abs/1901.02985)) we ran their standard search procedure on PSICOV and the lowest-resolution Darcy Flow task separately; we then evaluated the respective discovered architectures on PSICOV and all resolutions of Darcy Flow, applying the optimizer suggested by Auto-DeepLab but tuning the learning rate. We present the results here by linking two plots: a Darcy Flow plot ([link](https://drive.google.com/file/d/1t-rK4psYah2uWcuHeypx_NRc172OsZbg/view?usp=sharing)) to replace Figure 3 (left) and a PSICOV ([link](https://drive.google.com/file/d/1NPTwxhW021AMQXSoNnx3jiu55rHm-x9Y/view?usp=sharing)) to replace Figure 4. In both cases the DARTS operations tend to underperform XD; for Darcy Flow the result is even worse than the vanilla CNN, while for PSICOV DARTS matches XD when using shallow backbones but is worse when using more layers. XD also significantly outperforms Auto-DeepLab in both cases, although the latter does discover an architecture that performs consistently well across resolutions, unlike vanilla CNNs. These results demonstrate the utility of an operation space such as ours and will reinforce the submission-time results.
>
> 3\. [*One question I have is about how useful the resulting architectures are. In Darts, one can discretize the discrete search space resulting in very efficient architectures for future tasks. In this work, this doesn't seem so straight forward (and it is even stated that this was not done on purpose). Do the K-matrix style formulation lead to an efficient formula? Naively it seems like without it it would be up to 10x more costly (judging by the resnet-20 cifar10 walltimes).*]
> - You are correct that there is no discretization to an efficient architecture in this work and that ResNet-20 + XD is roughly 9x more expensive to train on CIFAR than a regular ResNet-20 (but only 4x more expensive at inference time). However, this multiplier is backbone-dependent; for example, LeNet and WideResNet-40-4 are only 3x more expensive to train (and 2x and 5x more expensive at inference-time, respectively). Furthermore, WideResNet-40-4 + XD is twice as fast to train as the full DARTS cell, albeit the latter must be trained for 3 times more epochs. Thus we believe a useful cost-benefit tradeoff can be found for datasets on which XD is better. Note that the time results for WideResNet and the DARTS cell are in Table 6 of the Appendix, and we also include more training time, memory, and parameter-count details in Table 5 of the Appendix (although note there are some typos that we have fixed in revision: CIFAR-10 x Conv should be 1.4 min/epoch, CIFAR-10 x XD should be 4.3 min/epoch, and Darcy Flow x Conv should be 0.028 min/epoch). To these we will add the inference-time results. Finally, note that the K-matrix software that we rely on is very recent, so there may be room for improvements in implementation efficiency.
>
> 4\. [*Related, but do you have any understanding of what kinds of "architectures" these things learn? Is anything interpretable?*]
> - This is a good question; it is indeed difficult to say exactly what architectures are being discovered because of the continuous nature of the architecture parameters. Nevertheless, we do make an attempt to understand them in Figure 2, where we show that operations discovered on permuted CIFAR data are further away from regular convolutions than those discovered on regular CIFAR data. The metric here is the Euclidean distance of the architecture parameters.

---

> > ### Comment · Reviewer_misB · 2021-08-20
> > **Response**
> >
> > Thank you for your thoughtful response. The new baselines are appreciated.
> >
> > In terms of a practical method, point 3 seems like a big disadvantage. Personally I would appreciate a line in the paper to this point contrasting it with, say, darts, or other methods that give a sparse network out.
> >
> > I plan to increase my score by 1, but plan to keep my confidence level.

---

> > > ### Author Response · Authors · 2021-08-20
> > > **Follow-up response to Reviewer misB**
> > >
> > > Thank you for your evaluation and feedback. We agree that efficiency at evaluation-time is an important direction for improvement. In revision we will add the requested note on the lack of sparse discretization and comparison to standard methods to the discussion at the end of Section 2.

---

### Decision · Program_Chairs · 2021-09-27

**Decision:**

Accept (Poster)

**Comment:**

This work is a well-presented and motivated perspective on NAS, where a reparameterisation of the search space using K-matrices yields a method that is able to operate across a number of interesting domains. The only major concern raised by the reviewers is, naturally, the choice of domains. The reviewers all agreed the authors have selected a diverse range of tasks and demonstrated their method on each. Naturally, given this demonstration, there is some anticipation of demonstrating this method on a diverse set of larger-scale problems, perhaps this will be future work.